# MINT-CoT: Enabling Interleaved Visual Tokens in Mathematical Chain-of-Thought Reasoning

**Xinyan Chen**[*1], **Renrui Zhang**[*†‡1], **Dongzhi Jiang**[1], **Aojun Zhou**[1]
**Shilin Yan, Weifeng Lin**[1], **Hongsheng Li**[‡1,2,3]

[1]CUHK MMLab  [2]Shanghai AI Laboratory  [3]CPII under InnoHK

{chenxyxy06, 1700012927zrr}@gmail.com
hsli@ee.cuhk.edu.hk

[*]Equal Contribution    [†]Project Leader    [‡]Corresponding Author

## Abstract

Chain-of-Thought (CoT) has widely enhanced mathematical reasoning in Large Language Models (LLMs), but it still remains challenging for extending it to multimodal domains. Existing works either adopt a similar textual reasoning for image input, or seek to interleave visual signals into mathematical CoT. However, they face three key limitations for math problem-solving: *reliance on coarse-grained box-shaped image regions, limited perception of vision encoders on math content, and dependence on external capabilities for visual modification.* In this paper, we propose **MINT-CoT**, introducing **M**athematical **IN**terleaved **T**okens for **C**hain-**o**f-**T**hought visual reasoning. MINT-CoT adaptively interleaves relevant visual tokens into textual reasoning steps via an Interleave Token, which dynamically selects visual regions of any shapes within math figures. To empower this capability, we construct the MINT-CoT dataset, containing 54K mathematical problems aligning each reasoning step with visual regions at the token level, accompanied by a rigorous data generation pipeline. We further present a three-stage MINT-CoT training strategy, progressively combining text-only CoT SFT, interleaved CoT SFT, and interleaved CoT RL, which derives our MINT-CoT-7B model. Extensive experiments demonstrate the effectiveness of our method for effective visual interleaved reasoning in mathematical domains, where MINT-CoT-7B outperforms the baseline model by +34.08% on MathVista, +28.78% on GeoQA, and +23.2% on MMStar, respectively. Our code and data are available at https://github.com/xinyan-cxy/MINT-CoT.

## 1  Introduction

Chain-of-Thought (CoT) [66, 32] has emerged as an effective strategy for enhancing the reasoning capabilities of Large Language Models (LLMs) [49, 51, 62, 72, 81, 39] by generating sequential rationales in their responses. In Multimodal Large Language Models (MLLMs) [50, 33, 88, 18, 20], CoT also plays a significant role [84] across various tasks involving image [41, 86, 40, 22, 17, 25], video [38, 4, 71, 14], and 3D [70, 24, 58, 21]. It enables MLLMs to reason over both textual and visual inputs, serving as a bridge that connects visual perception with abstract reasoning tasks.

However, despite these advances, applying CoT in mathematical reasoning with visual contexts remains challenging. Existing MLLMs mainly generate text-only reasoning steps for multimodal math problems [84, 85, 60, 79], simply adopting similar textual reasoning for image input. Nevertheless,

39th Conference on Neural Information Processing Systems (NeurIPS 2025).

**Question:**

In the given diagram, circle O has line segment AB as its diameter and CD as a chord. A tangent passing through point C intersects the extension of AB at point E, and angle E measures 42°. What is the measure of angle CDB? Choices: A: 22° B: 24° C: 28° D: 48°

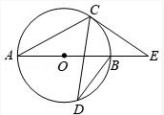

**Text-only CoT Reasoning:**

Step 1: Since angle E = 42°, therefore angle A = 42°. ✗

Step 2: Since AB is the diameter of circle O, angle ACB = 90°. Therefore, angle B = 180° - 42° - 90° = 48°.

Step 3: Since AB is the diameter of circle O, angle CDB = angle B = 48°. ✗

Answer: D ✗

> Lack of visual information leads to perception error.

**Box-shaped Visual CoT Reasoning:**

Step 1: Since angle E = 42°, angle CEB = 42° ✓

Step 2: Since AB is the diameter of circle O, angle ACB = 90°. ✓

Step 3: Therefore, angle CDB = angle CEB = 42°. ✗

Answer: D ✗

> Box-level cues are too coarse for math concepts.

**Visual Interleaved CoT Reasoning (Ours):**

Step 1: Connect OC, OC⊥CE. ✓

Step 2: AngleCOE = 180° - 90° - 42° = 48°. ✓

Step 3: Since OC = OD, angleCDB = angleODC = 1/2 angleBOC = 24 ✓

Answer: B ✓

> Token-level interleaved CoT is fine-grained and effective.

Figure 1: Comparison of three CoT reasoning methods: text-only CoT reasoning, box-shaped CoT reasoning and our visual interleaved CoT reasoning methods. (1) Text-only CoT lacks visual information, causing perception errors in mathematical reasoning. (2) Box-level cues are too coarse to capture complex visual structures in mathematical images. (3) Token-level interleaved CoT accurately identifies fine-grained visual regions to support reasoning.

due to the limited capability in perceiving math images, this strategy often fails to accurately interpret visual information within the CoT process, leading to reasoning errors.

Recent approaches have attempted to interleave visual content within reasoning steps through mechanisms such as bounding box selection and image cropping [55, 26, 76]. While effective in general visual scenarios, these methods still face three key limitations when extended to multimodal mathematical reasoning:

1. *Reliance on coarse-grained box-shaped image regions:* Recent advances introduce visual information into the CoT process by selecting image regions through bounding box-based methods. Visual-CoT [55], Visual SKETCHPAD [26], and VPT [76] all operate on box-shaped image regions, employing strategies such as bounding box generation, iterative masking, cropping, or re-encoding. However, as shown in Figure 1, these approaches all rely on bounding box-based cropping. While such box-level cues are effective in domains like object detection, where objects are typically isolated, they are too coarse-grained to capture the complex structures in mathematical images, where visual information is not discrete but highly interconnected. As a result, box-shaped selection tends to interleave too many irrelevant or misleading visual tokens, impairing the accuracy of mathematical reasoning.

2. *Limited perception of vision encoders on math content:* Some methods, like ICoT [16], adopt attention-based token selection to identify relevant visual tokens during reasoning without requiring additional training. These approaches rely heavily on visual features extracted by the vanilla vision encoders without specific tuning. However, as noted in MAVIS [83], mainstream vision encoders, which are primarily based on CLIP [54] or SigLIP [78], are pre-trained on natural images with general scenes, making mathematical images out-of-distribution. As a result, such methods often struggle to accurately locate relevant visual regions in complex mathematical tasks.

3. *Dependence on external capabilities for visual modification:* Other approaches attempt to enhance visual reasoning by dynamically generating new visual content or modifying existing images. MVoT [36] is built upon a unified autoregressive MLLM [59] to generate images as part of the CoT process, but it is only applicable to spatial planning tasks. Meanwhile, Visual SKETCHPAD requires external tools to draw on the original image in geometry-related tasks. These approaches depend on external capabilities, either requiring large-scale data to train the understanding model for generation, or relying on external tools with additional inference over the modified images, which leads to numerous extra costs.

Therefore, to address these challenges, we aim to propose a fine-grained, efficient visual interleaved CoT method to enhance the mathematical reasoning capabilities of MLLMs. In this paper, we introduce **MINT-CoT**, an approach of **M**athematical **IN**terleaved **T**oken selection for **C**hain-**o**f-**T**hought reasoning, which facilitates multimodal reasoning by interleaving relevant visual regions within reasoning steps. At the core of the MINT-CoT is the Interleave Token, a special token generated through the next-token prediction process. During reasoning, MINT-CoT automatically identifies and incorporates the most relevant visual tokens from the original image at each reasoning step. This is achieved by computing similarity scores between the output hidden states of the Interleave Token and all visual tokens, in order to identify the tokens most relevant to the mathematical concept at the current step. These selected visual tokens are then dynamically integrated into the textual reasoning steps, enabling the flexible selection of visual regions throughout the CoT process. In this way, the interleaved regions of mathematical images are not restricted to box-shaped areas but can flexibly include geometric shapes, line segments, coordinates, and other elements.

To enable effective training of MINT-CoT, we construct the MINT-CoT dataset, a 54K visual interleaved reasoning dataset. Each data point contains reasoning steps paired with the indices of selected tokens corresponding to the mathematical concepts involved in each step. We source mathematical problems from the Mulberry-260K dataset [74] to construct text-only CoT reasoning format, then annotate the reasoning steps with corresponding image regions through a four-step pipeline: (1) dividing images into grid-indexed regions, (2) mapping recognized text elements to grid indices via OCR-based text localization, (3) extracting key words, and (4) assigning visual regions to these key words using an advanced MLLM. This process creates a visual interleaved CoT reasoning dataset providing token-level supervision for training models to interleave visual content into reasoning steps.

Building on the MINT-CoT framework and MINT-CoT dataset, we design a progressive training strategy, the MINT-CoT training strategy, that incrementally improves MLLMs' ability with three training stages: (1) Text-only CoT Training, (2) Interleaved CoT SFT, and (3) Interleaved CoT RL. Through this training strategy, we train a MINT-CoT-7B model with the capability of mathematical visual interleaved CoT reasoning. Extensive experiments demonstrate the superiority of our proposed approach. Specifically, our method achieves absolute improvement of +32.59% on MathVista [43], +26.92% on GeoQA [5], and +23.2% on MMStar [7] benchmark compared to the baseline model.

Our main contributions are as follows:

- We propose MINT-CoT, which uses the Interleave Token to interleave fine-grained visual tokens within reasoning steps, enhancing multimodal mathematical reasoning.

- We construct the MINT-CoT dataset, a 54K dataset for multimodal mathematical reasoning, offering fine-grained alignment between textual rationales and visual inputs. We develop an automated pipeline to generate visual interleaved CoT data annotated with token indices.

- We develop a progressive three-stage MINT-CoT training strategy, to improve interleaved mathematical reasoning. Extensive experiments validate the efficiency of our method.

## 2 Related work

**MLLMs for Mathematics.** Recent advancements in MLLMs [50, 41, 2, 31] have shown impressive capabilities in various vision-language tasks. However, even powerful models like GPT-4V [50] and Qwen2-VL [63] fail to demonstrate satisfying performance on existing visual mathematical benchmarks [5, 44, 43], as highlighted by MathVerse [82]. Various specialized approaches [15, 83, 28, 9, 45, 57, 53] have emerged to enhance visual mathematical reasoning. Current approaches mostly focus on enriching the multimodal math data. G-LLaVA [15] extends the LLaVA architecture

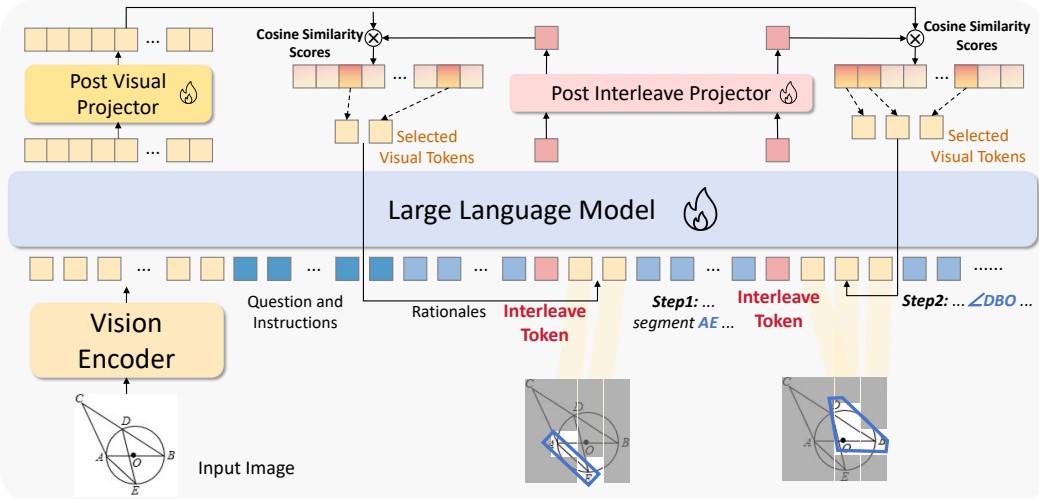

Figure 2: **Overview of the MINT-CoT framework.** During CoT reasoning, MINT-CoT generates an Interleave Token before each reasoning step and computes the similarity scores between embeddings projected by the decoder-side visual projector and the interleave projector. Based on these similarity scores, relevant visual tokens are selected, and the model inferences with these selected visual tokens.

with geometric reasoning capabilities by augmenting the current dataset. Math-LLaVA [57] enlarges the data scope with the introduced MathV360K dataset. MAVIS [83] first identifies the critical issue of the vision encoder and empowers it with the mathematical capability. Then it further develops an automated system for generating mathematical visual datasets at scale. Reverse Chain-of-Thought (R-CoT) [9] introduces the Geometry Generation Chain for creating geometric images with more accurate descriptions.

**Visual Chain of Thought.** With advancements of various visual reasoning tasks [43, 77, 30, 75, 69], visual chain of thought has been emerging as an effective method for both image generation [23, 29, 61, 87] and understanding [52, 74, 60] tasks. Our work focuses on leveraging it for reasoning on images, where two distinct methods have emerged. One line of the method relies on textual CoT to conduct multimodal analysis [11, 46, 6, 79, 10, 73]. For example, R1-V [6] extends the paradigm of DeepSeek R1 [19] to generate a comprehensive text CoT to analyze the visual information before providing the final answer. Another line of method explicitly incorporates multimodal elements in the rational [55, 47, 67, 26, 35]. Visual CoT [55] and Chain-of-Spot [42] propose to crop the region of high interest on the image and integrate it into the CoT process. Chain-of-Image [47] and Visual SKETCHPAD [26] introduce auxiliary tools to generate helpful diagrams for mathematical or geometric problem-solving. Although these methods demonstrate competitive performance, they are limited to rigid image cropping or dependence on external tools. Recently, ICoT [16] leverages the attention map of the MLLM to select the relevant visual tokens to compose the multimodal rational. However, this approach relies solely on attention scores on the image feature maps, which have been shown to be insufficiently informative for mathematical scenarios [83].

## 3 Method

To address the challenges of multimodal CoT in mathematical reasoning, we propose MINT-CoT. In this section, we first introduce the framework of MINT-CoT in Section 3.1. Then we introduce the MINT-CoT dataset and provide a detailed discussion of the dataset generation method in Section 3.2. Finally, we present the progressive MINT-CoT training strategy in Section 3.3.

### 3.1 MINT-CoT

Previous CoT approaches in MLLMs mainly generate text-based reasoning steps, which are not explicitly grounded in visual features and therefore struggle with mathematical reasoning that involves

visual details. We formulate this CoT reasoning process as:

$$\{s^{(1)}, s^{(2)}, \ldots, s^{(k)}\}, answer = \text{LLM}(V, \text{TextEncoder}(T)), \qquad (1)$$

where $V = \text{VisionEncoder}(I) = \{v_\tau\}_{\tau=1}^{N}$ denotes the visual feature extracted from the input image $I$, and each $v_\tau$ represents the $\tau$-th visual token generated by the vision encoder. $T$ denotes the input mathematical question and instructions, $\{s^{(i)}\}$ is the sequence of textual reasoning steps generated by the model, and $answer$ is the final answer. Recent advancements attempt to incorporate multimodal reasoning steps in the CoT process. However, current coarse-grained methods only focus on selecting box-shaped visual regions; how to adaptively select the visual content in alignment with each textual reasoning step remains an open question. We thus propose the **MINT-CoT** framework and introduce an Interleave Token to help MLLMs select visual tokens from the visual feature $V$. The overview of the MINT-CoT framework is illustrated in Figure 2.

**Interleave Token.** An Interleave Token is a special token generated prior to each reasoning step. It is used to select visual tokens that are relevant to the mathematical concepts involved in that step (e.g., "line segment AB", "angle DOC"), thereby facilitating the reasoning process. When an Interleave Token is output in step $i$, its output hidden state $h_{\text{post\_intlv}}^{(i)}$ is projected via a post interleave projector $P_{\text{post\_intlv}}$, while all the output hidden states of the visual tokens $h_{\text{post\_vis}}$ are projected via a post visual projector $P_{\text{post\_vis}}$. The cosine similarity between the two projected embeddings is first computed and then scaled by a learnable parameter $\gamma$:

$$\alpha^{(i)} = \gamma \cdot \cos\left(P_{\text{post\_intlv}}(h_{\text{post\_intlv}}^{(i)}), \; P_{\text{post\_vis}}(h_{\text{post\_vis}})\right). \qquad (2)$$

Each tokens' similarity score $\alpha_\tau^{(i)}$ is then compared against a predefined threshold $\theta$, and visual tokens with scores above this threshold are selected:

$$\{v^{(i)}\} = \{v_\tau^{(i)} \mid \alpha_\tau^{(i)} > \theta\}. \qquad (3)$$

The selected tokens $\{v^{(i)}\}$ are interleaved into the reasoning process at step i. In this way, the important visual regions are interleaved into the model, prior to each textual step, enhancing visual perception and improving reasoning accuracy.

**Inference with Interleaved Visual Tokens.** With the selected visual tokens $\{v^{(i)}\}$ obtained at each reasoning step, MINT-CoT interleaves both visual content and text-based reasoning steps throughout the inference process, ultimately producing the final answer. Formally, this process extends the standard CoT formulation in Eq. 1 as:

$$\{v^{(1)}, s^{(1)}, v^{(2)}, s^{(2)}, \ldots, v^{(k)}, s^{(k)}\}, answer = \text{LLM}(V, \text{TextEncoder}(T)). \qquad (4)$$

This interleaved token selection mechanism enables the model to explicitly ground visual evidence throughout the reasoning chain, thereby facilitating visual interleaved CoT reasoning for solving multimodal mathematical problems.

### 3.2 Dataset Curation

To empower MINT-CoT capabilities for MLLMs, we develop a data generation pipeline that automatically generates mathematical visual interleaved data annotated with selected token indices, and obtain 54K samples for model training. To construct the text-only cot format of our dataset, we begin by selecting mathematical problems from the Mulberry-260K dataset [74], which was created using Collective Monte Carlo Tree Search and demonstrates strong performance on reasoning tasks. Specifically, we extract the *"### Rationale"* and *"### Steps"* sections from the dataset as the reference reasoning steps for our task. Using these sections alongside the corresponding images, we follow a four-step data construction process, as shown in Figure 3:

1. **Grid Images.** To obtain the indices of visual tokens for subsequent token index annotation in textual reasoning steps, we divide the original images into grid cells. Following the patch-splitting strategy used in vision encoders such as Vision Transformer [12], each image is partitioned into a grid, and a unique index is assigned to each cell. These grid cells and their indices are subsequently overlaid onto the original images to produce *grid-indexed images*.

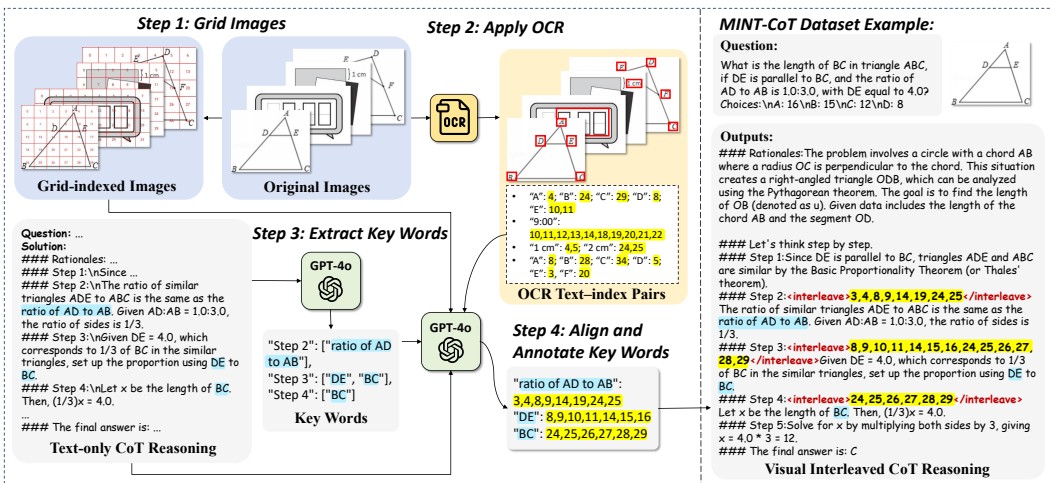

Figure 3: **Data generation pipline.** *Step 1: Grid Images.* We divide each image into grid cells and assign index values to each cell. *Step 2: Apply OCR.* We use PaddleOCR to recognize textual elements and associate them with corresponding grid indices. *Step 3:* Extract Key Words. We employ GPT-4o to extract key words from each reasoning step. *Step 4: Align and Annotate Key Words.* We use GPT-4o to annotate each key word with the grid indices, and get the final visual interleaved CoT reasoning steps.

2. **Apply OCR.** Then, to more accurately map token indices onto textual reasoning steps, we apply PaddleOCR [37] to recognize textual elements in the original images. And we align the bounding boxes of the detected text with their corresponding grid indices, thereby constructing *"OCR text–index" pairs*.

3. **Extract Key Words.** Certain mathematical concepts often play a significant role in each reasoning step. Selecting visual tokens closely related to these concepts can improve reasoning accuracy. Therefore, we employ GPT-4o [12] to extract *key words* from each reasoning step. Since the extracted key words are used in the subsequent annotation with visual indices, they are extracted only when a reasoning step contains links to visual tokens.

4. **Align and Annotate Key Words.** Finally, given the grid-indexed images, the "### Rationale" and "### Steps" sections, the "OCR text–index" pairs, and the extracted key words, we prompt GPT-4o to annotate each key word with the corresponding grid indices. These *annotated indices* are subsequently inserted into the reasoning steps associated with their corresponding key words, resulting in a visual-interleaved CoT reasoning dataset.

Through this process, we construct a dataset of 54K samples, where the reasoning steps are annotated with corresponding grid indices. As shown in the right column of Figure 3, each data point consists of a mathematical problem and an image as input, with the corresponding visual interleaved CoT response as output. This dataset serves as the foundation for training the MINT-CoT models. Further details are provided in Appendix A.2.

## 3.3 Training strategy

Building on the previously introduced MINT-CoT framework and dataset, we now describe the corresponding MINT-CoT training strategy, which consists of three stages: (1) Text-only CoT Training, (2) Interleaved CoT SFT, and (3) Interleaved CoT RL.

**Stage 1: Text-only CoT SFT.** To enable the MLLM to adopt a general reasoning format, we first train the base model using the text-only CoT reasoning data in MINT-CoT dataset, without visual interleaving. This stage serves as a foundation for subsequent interleaved training.

**Stage 2: Interleaved CoT SFT.** In the second stage, we aim to train the model to select visual tokens using the Interleave Token and adapt to reasoning with interleaved visual content. The model

is fine-tuned with a loss that jointly optimizes both textual reasoning and visual alignment. As introduced in Eq. 4, the output sequence of MINT-CoT alternates between sets of selected visual tokens $v^{(i)}$ and textual reasoning steps $s^{(i)}$, followed by the final answer:

$$\{v^{(1)}, s^{(1)}, v^{(2)}, s^{(2)}, \dots, v^{(k)}, s^{(k)}\}, \text{answer} \sim P_\theta(\cdot \mid I, T), \tag{5}$$

We first apply a cross-entropy loss to textual tokens at positions $\mathbf{T} \subset \{1, 2, \dots, T\}$ covering all segments $\{s^{(i)}\}$ and the answer, while conditioning on the full preceding sequence. Let $Y = \{y_1, y_2, \dots, y_T\}$ denotes the full sequence of output tokens. Specifically, the loss for predicting the next textual token is defined as:

$$\mathcal{L}_{\text{CE}} = -\sum_{t \in \mathbf{T}} \log P_\theta\big(y_t \mid y_{<t}, I, T\big) \tag{6}$$

We do not supervise the cross-entropy loss for predicting the Interleave token. Instead, we manually concatenate it at each step, and during inference, we concatenate the Interleave Token whenever the "*### Step*" marker is generated. To supervise the interleaved visual tokens, we apply a binary cross-entropy loss on the scaled cosine similarity scores $\alpha$ introduced in Eq. 2 with ground-truth labels $X \in \{0, 1\}$:

$$\mathcal{L}_{\text{BCE}} = -\sum_{i=1}^{N} \sum_{j=1}^{L} \Big( X_{ij} \log \sigma(\alpha_{ij}) + (1 - X_{ij}) \log(1 - \sigma(\alpha_{ij})) \Big), \tag{7}$$

where $N$ is the number of Interleaved Tokens in a batch, $L$ is the length of input visual tokens, and $\sigma(\cdot)$ denotes the sigmoid function. The final training objective is defined as the sum of both losses:

$$\mathcal{L} = \mathcal{L}_{\text{CE}} + \mathcal{L}_{\text{BCE}}. \tag{8}$$

This combined loss guides the model to jointly align visual tokens and perform interleaved reasoning.

**Stage 3: Interleaved CoT RL.** To move beyond supervised annotations, we aim to enable the model to autonomously explore more flexible and effective selection of visual tokens guided by reasoning objectives, and enhance its ability to perform interleaving CoT reasoning. Reinforcement learning provides a natural framework for this goal. To this end, we extend the Group Relative Policy Optimization (GRPO) [56] framework to our MINT-CoT training strategy. For a group of reasoning chains with group size $G$, we compute answer correctness as the reward $r \in \{0, 1\}$ and define the advantage via group-wise comparison as $\hat{A}_j = \frac{r_j - \text{mean}(\mathbf{r})}{\text{std}(\mathbf{r})}$, where $r_j$ indicates if the $j$-th chain of steps in a group yields the correct answer. The policy loss for the generated tokens is then formulated as:

$$\mathcal{L}_{\text{GRPO}} = -\mathbb{E}_{\{Y_j\}_{j=1}^{G}} \left[ \frac{1}{G} \sum_{j=1}^{G} \left( \frac{P_\theta(Y_j)}{P_{\theta_{\text{old}}}(Y_j)} \hat{A}_j - \beta D_{\text{KL}}[P_\theta \parallel P_{\text{ref}}] \right) \right], \tag{9}$$

where $P_{\text{ref}}$ is a reference policy that serves as a regularization target. This stage further strengthens the model's reasoning ability with visual interleaved content, ultimately resulting in MINT-CoT-7B. Additional theoretical details of this training stage are provided in Appendix A.3.

# 4 Experiments

In this section, we first introduce the experimental settings in Section 4.1. Then, we discuss the quantitative results and ablation study in Section 4.2 and Section 4.3 respectively. Finally, we present the qualitative results in Section 4.4.

## 4.1 Experimental Settings

**Implementation Details.** We build on Qwen2-VL-7B [64] and train our model in three stages with a combination of SFT and RL on the MINT-CoT dataset. All model parameters except the vision encoder are updated. Full implementation details are provided in Appendix A.4.

**Test Benchmark.** We evaluate MINT-CoT on three mathematical benchmarks: GeoQA [5], Math-Vista [43] and MMStar [7]. GeoQA is a benchmark of geometric problems with annotated solution programs. To evaluate on GeoQA, we follow R1-V [6] and Hint-GRPO [27] using the Geo170K test set [15], the English version of the GeoQA benchmark. MathVista is a benchmark designed to

Table 1: **Combined quantitative results on MathVista.** We evaluate MINT-CoT-7B, the baseline model, and state-of-the-art general and reasoning MLLMs on the mathematical subset of MathVista. MINT-CoT significantly outperforms the baseline model and achieves superior performance compared to open-source reasoning models. **Bold** and underlined results indicate the best and second-best among open-source models, respectively.

| Model | #Params | MathVista-Math | | | | |
| --- | --- | --- | --- | --- | --- | --- |
| | | All | GEO | ALG | GPS | TQA |
| *Closed-Source Model* | | | | | | |
| GPT-4o [48] | – | 66.67 | 63.68 | 67.04 | 63.46 | 77.42 |
| Claude-3.5 Sonnet [1] | – | 67.41 | 65.09 | 67.79 | 65.38 | 74.19 |
| *Open-Source General Model* | | | | | | |
| LLaVA-OneVision-Qwen2-7b-ov [34] | 7B | 67.04 | 69.34 | 67.04 | 69.71 | 58.06 |
| InternVL2-8B [8] | 8B | 62.59 | 62.26 | 62.92 | 62.50 | 62.90 |
| InternVL2-8B-MPO [65] | 8B | 68.52 | 68.87 | 68.91 | 69.71 | 64.52 |
| DeepSeek-VL2 [68] | 4.5B | 65.56 | 63.68 | 65.54 | 63.94 | 70.97 |
| Qwen2.5-VL-7B-Instruct [3] | 7B | 66.66 | 65.56 | 66.29 | 65.87 | 69.35 |
| *Open-Source Reasoning Model* | | | | | | |
| Open-R1-Multimodal [13] | 7B | 54.81 | 52.36 | 54.68 | 53.37 | 59.68 |
| R1-VL-7B [80] | 7B | 69.63 | 68.87 | 69.66 | 69.71 | 69.35 |
| Mulberry [74] | 7B | 68.52 | 67.92 | 68.54 | 68.75 | 67.74 |
| MM-Eureka [46] | 7B | 72.59 | 71.22 | 72.66 | 72.60 | **72.58** |
| Qwen2-VL-7B-Instruct [64] (Baseline) | 7B | 41.11 | 35.85 | 41.57 | 36.54 | 56.45 |
| MINT-CoT-7B | 7B | **73.70** | **74.53** | **73.78** | **75.00** | 69.35 |
| $\Delta$ *over the Baseline Model* | | *+32.59* | *+38.63* | *+32.21* | *+38.46* | *+12.9* |

Table 2: **Combined quantitative results of on GeoQA.** We evaluate MINT-CoT-7B, the baseline model and the state-of-the-arts.

| Model | GeoQA |
| --- | --- |
| Qwen2.5-VL-7B-Instruct [3] | 43.50 |
| R1-V [6] | 59.00 |
| Open-R1-Multimodal [13] | 48.67 |
| Hint-GRPO [27] | 55.31 |
| Qwen2-VL-7B-Instruct [64] (Baseline) | 37.80 |
| MINT-CoT-7B | **64.72** |
| $\Delta$ *over the Baseline Model* | *+26.92* |

Table 3: **Combined results on the mathematical subset of MMStar.** We evaluate MINT-CoT-7B, the baseline model and the state-of-the-arts.

| Model | MMStar-Math |
| --- | --- |
| Qwen2.5-VL-7B-Instruct [3] | 66.8 |
| InternVL2-8B [8] | 66.8 |
| R1-VL-7B [79] | 68.4 |
| Mulberry [74] | 66.8 |
| Open-R1-Multimodal [13] | 59.2 |
| Qwen2-VL-7B-Instruct [64] (Baseline) | 46.4 |
| MINT-CoT-7B | **69.6** |
| $\Delta$ *over the Baseline Model* | *+23.2* |

integrate challenges from diverse mathematical and visual tasks. As our paper targets specifically mathematical problems, we extract the mathematical subsets (FunctionQA, Geometry3K, GeoQA+, GEOS, and UniGeo), i.e., 'MathVista-Math' in Table 1, and report accuracy scores across four primary tasks: geometry reasoning (GEO), algebraic reasoning (ALG), geometry problem solving (GPS), and textbook question answering (TQA). MMStar is a multi-modal benchmark covering different core capabilities and detailed axes. For evaluation, we also extract the mathematical capability dimension, referred to as "MMStar-Math".

## 4.2 Quantitative Results

**Comparison with the Baseline.** As shown in Table 1 for the results of mathematical subsets of MathVista, our MINT-CoT-7B achieves an improvement of up to +32.59% over the baseline, and improves a lot on all four primary tasks. This strongly demonstrates the effectiveness of our MINT-CoT framework and training strategy. Table 2 presents the results on the GeoQA benchmark, where our MINT-CoT-7B outperforms the baseline model by +26.92%. Similarly, in Table 3, MINT-CoT-7B outperforms the baseline model by +23.2% on MMStar-Math, validating the efficiency of MINT-CoT on geometry problems.

Table 4: **Ablation study on different training stages.** We evaluate the three progressive training stages on different benchmarks.

| Model | MMStar-Math | GeoQA | MathVista-Math | | | | |
|---|---|---|---|---|---|---|---|
| | | | All | GEO | ALG | GPS | TQA |
| Baseline [64] | 46.4 | 37.80 | 41.11 | 35.85 | 41.57 | 36.54 | 56.45 |
| + Text-only CoT SFT | 67.6 | 59.02 | 64.07 | 64.15 | 64.04 | 64.42 | 62.90 |
| + Interleaved CoT SFT | 68.0 | 62.07 | 67.78 | 66.51 | 67.79 | 67.31 | **69.35** |
| + Interleaved CoT RL | **69.6** | **64.72** | **73.70** | **74.53** | **73.78** | **75.00** | **69.35** |

Table 5: **Ablation study of different interleaving methods on GeoQA and MathVista-Math.** Our Interleaved CoT SFT achieves the highest improvement on both benchmarks, demonstrating the effectiveness of our interleaved token selection method.

Figure 4: F1 score plot of visual token selection during Interleaved CoT SFT.

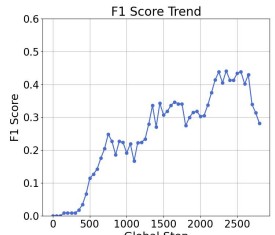

| Model | GeoQA | MathVista-Math | | | | |
|---|---|---|---|---|---|---|
| | | All | GEO | ALG | GPS | TQA |
| Original | 37.80 | 41.11 | 35.85 | 41.57 | 36.54 | 56.45 |
| Text-only CoT SFT | 59.02 | 64.07 | 64.15 | 64.04 | 64.42 | 62.90 |
| Original Image CoT SFT | 61.41 | 40.37 | 38.68 | 40.82 | 39.42 | 43.54 |
| Bounding Box CoT SFT | 61.80 | 65.56 | 63.21 | 65.54 | 63.94 | **70.97** |
| Interleaved CoT SFT (Ours) | **62.07** | **67.78** | **66.51** | **67.79** | **67.31** | 69.35 |

**Comparison with State-of-the-arts.** We also compare our model with state-of-the-art MLLMs, including closed-source model, open-source models, and open-source reasoning models. Specifically, for open-source reasoning models, we choose recent works like R1-VL-7B [79], MM-Eureka [46] and Open-R1-Multimodal [13]. As shown in Table 1, our model achieves the highest overall accuracy on the MathVista mathematical subsets, outperforming both open-source reasoning models and general models, and surpassing the best-performing open-source MLLM by +1.11% as well as closed-source models, demonstrating strong capabilities in mathematical reasoning. On geometry reasoning, geometry problem solving and algebraic reasoning, MINT-CoT-7B outperforms state-of-the-art models by +3.31%, +1.12%, and +2.4%, respectively. However, for textbook question answering, our performance is slightly below MM-Eureka. On the GeoQA benchmark, as shown in Table 2, our model outperforms the state-of-the-art models by +5.72%. In Table 3, MINT-CoT-7B also outperforms the state-of-the-art by +1.2% on MMStar-Math, further demonstrating its capability in geometry reasoning.

### 4.3 Ablation Study

**Training Stage Ablation.** We conduct an ablation study on the different training stages of MINT-CoT, as described in Section 3.3. The results on different benchmarks are presented in Table 4. The Text-only CoT SFT stage improves performance by +21.2% on MMStar-Math, +21.22% on GeoQA, and +22.96% on MathVista-Math, as it helps the model learn the general reasoning format illustrated in the left column of Figure 3. The Interleaved CoT SFT stage further boosts performance by +0.4% on MMStar-Math, +3.05% on GeoQA, and +3.71% on MathVista-Math across all primary tasks by enabling the model to interleave visual tokens into textual reasoning steps. Finally, the Interleaved CoT RL stage enhances performance by an additional +1.6% on MMStar-Math, +2.65% on GeoQA, and +5.92% on MathVista-Math through reinforcement learning, which enables the model to reason more effectively with interleaved tokens.

**Interleaving Method Ablation.** We conduct an ablation study on the interleaving method used in the Interleaved CoT SFT stage, with the results presented in Table 5. Starting with the model trained in the Text-only CoT SFT stage, we simply interleave the original image into each reasoning step without the use of projectors or the Interleave token structure, which we refer to as "Original Image CoT SFT". We find that, on MathVista-Math, the performance of Original Image CoT SFT significantly decreases compared to Text-only CoT SFT. On the GeoQA benchmark, it also underperforms our Interleaved CoT SFT. This decline is likely due to the interleaving of excessive unrelated visual tokens during reasoning. Furthermore, we train a model that uses the Interleave token to select a

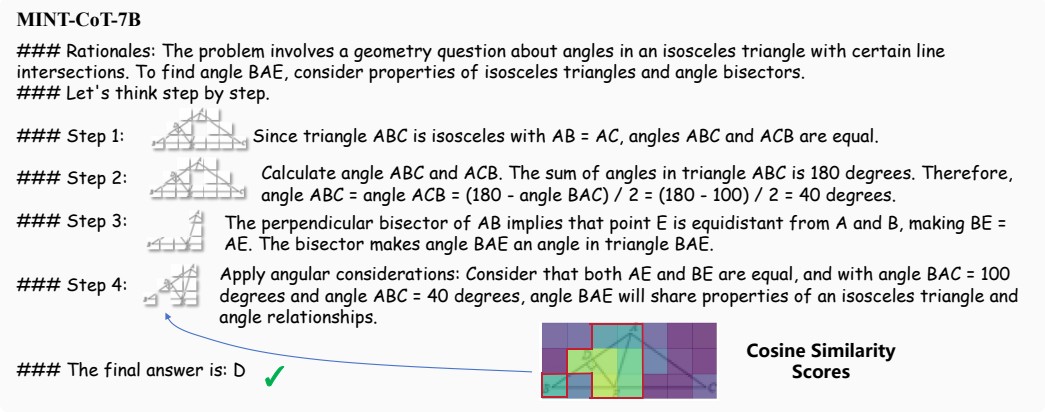

**Question:**
Given triangle ABC where AB = AC and angle BAC measures 100.0 degrees, and the perpendicular bisector of AB intersects AB and BC at points D and E, what is the measure of angle BAE? Represent the measure of angle BAE as θ. Choices: A: 80° B: 60° C: 50° D: 40°

**Qwen2vl-7B-Instruct**
To solve this problem, we first need to visualize the triangle and the given information:\n\n1. Triangle ABC is isosceles with AB = AC and angle BAC = 100.0 degrees.\n2. The perpendicular bisector of AB intersects AB at point D and intersects BC at point E.\n\nSince AB = AC, triangle ABC is isosceles as well. The perpendicular bisector of AB implies that AE is the median to the hypotenuse of triangle ABC (because D is the midpoint of AB). \n\nAlso, in an isosceles triangle, if a median is drawn to the hypotenuse, it creates two right-angled triangles. Let's denote the measure of angle BAE as θ. Due to the properties of the isosceles triangle and the perpendicular bisector, angle BAE (θ) should be half of the apex angle BAC.\n\nSince ∠BAC = 100.0 degrees, θ = 100.0 / 2 = 50.0 degrees.\n\nTherefore, the correct answer is C.  ✗

**MINT-CoT-7B**
### Rationales: The problem involves a geometry question about angles in an isosceles triangle with certain line intersections. To find angle BAE, consider properties of isosceles triangles and angle bisectors.
### Let's think step by step.

### Step 1: Since triangle ABC is isosceles with AB = AC, angles ABC and ACB are equal.

### Step 2: Calculate angle ABC and ACB. The sum of angles in triangle ABC is 180 degrees. Therefore, angle ABC = angle ACB = (180 - angle BAC) / 2 = (180 - 100) / 2 = 40 degrees.

### Step 3: The perpendicular bisector of AB implies that point E is equidistant from A and B, making BE = AE. The bisector makes angle BAE an angle in triangle BAE.

### Step 4: Apply angular considerations: Consider that both AE and BE are equal, and with angle BAC = 100 degrees and angle ABC = 40 degrees, angle BAE will share properties of an isosceles triangle and angle relationships.

### The final answer is: D  ✓

**Cosine Similarity Scores**

Figure 5: Qualitative results of Qwen2-VL-7B-Instruct and MINT-CoT-7B. MINT-CoT-7B demonstrates improved CoT reasoning capability by interleaving fine-grained visual tokens. There is also a visualization of the similarity scores for the Interleaved Token generated during Step 4.

rectangular region of visual tokens at each reasoning step, referred to as "Bounding Box CoT SFT". As shown in the table, this approach underperforms our Interleaved CoT SFT on both benchmarks, except for the TQA task, and even underperforms the Text-only CoT SFT on GEO and GPS tasks in MathVista-Math. These results demonstrate the effectiveness of our token selection method for mathematical reasoning tasks.

### 4.4 Qualitative Results

We present the qualitative results of the baseline model Qwen2-VL-7B-Instruct and our proposed model MINT-CoT-7B, as shown in Figure 5. Compared to the baseline, MINT-CoT-7B demonstrates a more coherent reasoning format and is capable of selecting and interleaving relevant visual tokens during inference. More qualitative results of our model are shown in Appendix A.6. Moreover, we provide a plot of the average F1 score between the selected visual tokens and ground truth visual tokens in each reasoning step during the Interleaved CoT SFT stage, as shown in Figure 4. For the Interleaved CoT RL stage, we do not report an F1 score plot due to the absence of ground truth visual token indices for online inference. As shown in the plot, the F1 score exhibits a fluctuating upward trend during training, demonstrating that the accuracy of visual token selection is increasing during the Interleaved CoT SFT training strategy.

## 5 Conclusion

In this paper, we first propose MINT-CoT, a method for enhancing multimodal mathematical reasoning by interleaving fine-grained visual tokens into CoT. We use the novel Interleave Token to automatically select visual tokens for each reasoning step. Then, we introduce the MINT-CoT dataset and a four-step dataset generation pipeline. Finally, we present the MINT-CoT training strategy, which includes Text-only CoT Training, Interleaved CoT SFT and Interleaved CoT RL, enhancing the MLLMs' ability to reason over interleaved visual tokens. Our experiments with the obtained MINT-CoT-7B model demonstrate significant improvements across various benchmarks.

## Acknowledgements

This study was supported in part by National Key R&D Program of China Project 2022ZD0161100, in part by the Centre for Perceptual and Interactive Intelligence, a CUHK-led InnoCentre under the InnoHK initiative of the Innovation and Technology Commission of the Hong Kong Special Administrative Region Government, in part by NSFC-RGC Project N_CUHK498/24, and in part by Guangdong Basic and Applied Basic Research Foundation (No. 2023B1515130008, XW).

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

# A   Appendix

## A.1   Overview

We organize our supplementary material as follows.

- Dataset Details
    - Dataset Example
    - Dataset Statistic
- Theoretical Details of Interleaved CoT RL
- Additional Implementation Details
- Additional Ablation Study
    - Text-only Method Ablation
    - Results on Other Benchmark
    - Projector Ablation
- Additional Qualitative Results
- Limitations

## A.2   Dataset Details

**Dataset Example**   We present examples from our MINT-CoT Dataset in Figures 6 to 8, where the yellow highlights indicate the interleaved grid indices, and the blue highlights denote the key words in each reasoning step.

**Dataset Statistic**   We provide the key statistics of MINT-CoT Dataset in Table 6. This dataset comprises 54,031 data points derived from the mathematical portion of the Mulberry-260k dataset.

Table 6: **Key statistics of the MINT-CoT dataset.**

| Statistic | Value |
|---|---|
| Total data points | 54,031 |
| Data points containing Interleave Tokens (interleaved data points) | 52,142 |
| Average number of Interleave Tokens per interleaved data point | 2.80 |
| Maximum number of Interleave Tokens in a single interleaved data point | 12 |
| Average number of selected indices per interleaved data point | 19.91 |
| Average number of selected indices per Interleave Token | 7.10 |
| Minimum number of selected indices in a single Interleave Token | 1 |
| Maximum number of selected indices in a single Interleave Token | 140 |

## A.3   Theoretical Details of Interleaved CoT RL

Following the standard GRPO framework [56], we integrate GRPO into our approach. Specifically, similar to $\mathcal{L}_{\text{CE}}$ in Stage 2, we apply a policy loss $\mathcal{L}_{\text{GRPO\_text}}$ to textual tokens:

$$\mathcal{L}_{\text{GRPO\_text}} = -\mathbb{E}_{\{Y_j\}_{j=1}^{G} \sim P_{\theta_{\text{old}}}(\cdot|I,T)} \left[ \frac{1}{G} \sum_{j=1}^{G} \frac{1}{|\mathbf{T}_j|} \sum_{t \in \mathbf{T}_j} \left\{ \frac{P_\theta(y_{j,t}|y_{j,<t},I,T)}{P_{\theta_{\text{old}}}(y_{j,t}|y_{j,<t},I,T)} \cdot \hat{A}_{j,t} - \beta D_{\text{KL}}[P_\theta \parallel P_{\text{ref}}] \right\} \right], \quad (10)$$

where $\hat{A}_{j,t}$ is the advantage detailed in Section 2.3, $P_{\text{ref}}$ is a reference policy that serves as a regularization target, and $D_{\text{KL}}[P_\theta \parallel P_{\text{ref}}]$ penalizes deviation from this reference distribution to encourage stable updates. The min and clip operations are omitted for brevity.

To enable more flexible and effective selection of visual tokens, we further apply a $\mathcal{L}_{\text{GRPO\_vis}}$ to the scaled similarity scores $\alpha_{j,\tau}^{(i)}$, which are derived from the interactions between Interleave tokens and input visual tokens in the the $j$-th chain of reasoning steps. Let $N_j$ denote the the number of

Table 7: **Ablation study on the post interleave projector and the post visual projector.** We compare three configurations: without projectors, with single-layer linear projections, and with two-layer MLPs.

| Configuration | Layer Number | All | GEO | ALG | GPS | TQA |
|---|---|---|---|---|---|---|
| w.o. projectors | – | 64.44 | 63.68 | 64.42 | 63.94 | 66.13 |
| w. projectors | 1 | **67.78** | **66.51** | **67.79** | **67.31** | **69.35** |
|  | 2 | 65.18 | 63.21 | 65.54 | 63.94 | **69.35** |

reasoning steps in $j$-th chain, and $M_j^{(i)}$ denote the number of visual tokens interleaved in the $i$-th reasoning step in the $j$-th chain. Formally, the loss is defined as:

$$\mathcal{L}_{\text{GRPO\_vis}} = -\mathbb{E}_{\{Y_j\}_{j=1}^G \sim P_{\theta_{\text{old}}}(\cdot|I,T)} \left[ \frac{1}{G} \sum_{j=1}^G \frac{1}{N_j} \sum_{i=1}^{N_j} \frac{1}{M_j^{(i)}} \sum_{\tau=1}^{M_j^{(i)}} \left\{ \frac{P_\theta(\alpha_{j,\tau}^{(i)}|y_{j,<\tau},I,T)}{P_{\theta_{\text{old}}}(\alpha_{j,\tau}^{(i)}|y_{j,<\tau},I,T)} \cdot \hat{A}_j - \beta D_{\text{KL}}[P_\theta \| P_{\text{ref}}] \right\} \right]. \quad (11)$$

The final policy loss is defined as the sum of both losses, with the $\mathcal{L}_{\text{GRPO\_vis}}$ rescaled by a weighting factor $\lambda$:

$$\mathcal{L}_{\text{GRPO}} = \mathcal{L}_{\text{GRPO\_text}} + \lambda \cdot \mathcal{L}_{\text{GRPO\_vis}}. \quad (12)$$

By computing this combined loss, we enhance both token selection and inference capabilities using Interleave tokens.

## A.4 Additional Implementation Details

We use Qwen2-VL-7B [64] as the base MLLM model in our experiments. Each of the two projectors, $P_{\text{interleave}}$ and $P_{\text{vis}}$, is implemented as a single linear layer. We uniformly set the threshold $\theta = 0.7$ to filter the similarity scores. The hyper-parameter $\gamma$ to scale the similarity is set to $1/0.07$ following CLIP [54]. The training procedure consists of three stages: *(1) Text-only CoT Training*, where we train for 2 epochs on the MINT-CoT dataset without applying the interleaving strategy, using a learning rate of 5.0e-6 and a batch size of 64, following the configuration of Mulberry [74]; *(2) Interleaved CoT SFT*, where we train for 3 epochs on the MINT-CoT dataset with a learning rate of 1e-6 and a batch size of 64; and *(3) Interleaved CoT RL*, where we train for 700 steps on the MINT-CoT dataset, using a group size $G = 4$, a weighting factor $\lambda = 0.02$, a learning rate of 1e-6 and a batch size of 16. During training, all model parameters, including the Interleave Token and projector layers, are unfrozen, except for the vision encoder, which remains fixed. Finally, the resulting model is named MINT-CoT-7B.

For Bounding Box CoT SFT, we use the MINT-COT dataset and extract the minimal enclosing rectangle that covers the index positions of all labels as the ground truth bounding box to train the model. We train 2 epochs with a learning rate of 1e-6 and a batch size of 64. And during inference, it interleave the minimal enclosing rectangle that covers all the seleted tokens. For Original Image CoT SFT, however, we enforce the concatenation of the entire image at the beginning of each step during both training and inference. We train only 1 epoch with a learning rate of 1e-6 and a batch size of 64,

## A.5 Additional Ablation Study

**Text-only Method Ablation** We conduct an ablation study using a text-only training pipeline. This involves first applying a Text-only CoT SFT stage to the baseline model, followed by a Text-only CoT RL stage using the GRPO framework. As shown in Table 8, MINT-CoT still outperforms this setting by 2.96% on the MathVista mathematical subsets, showcasing the superiority of our Interleaved CoT approach.

**Results on Other Benchmark** In addition to the quantitative results in Section 3.2, we provide results of the different training stages of our method on the vision-only input setting of the Mathematics section of MMMU-Pro [77] benchmark. As shown in Table 9, MINT-CoT-7B outperforms the baseline model by +5%. and the performance improves with the Interleaved CoT approach, validating the efficiency of our method.

Table 8: Ablation study of the text-only training method.

| Model | MathVista-Math |
|---|---|
| Baseline | 41.11 |
| + Text-only CoT SFT | 64.07 |
| + Text-only CoT RL | 70.74 |
| + Interleaved CoT SFT | 67.78 |
| + Interleaved CoT RL | **73.70** |

Table 9: Ablation study of different training stages on the Mathematics section of the MMMU-Pro benchmark.

| Model | MMMU-Pro-V |
|---|---|
| Baseline | 25 |
| + Text-only CoT SFT | 25 |
| + Interleaved CoT SFT | **30** |
| + Interleaved CoT RL | **30** |

**Question:**
The Municipal Planning Bureau is designing a cross-street bridge with a height of 'z' units, where z = 6.0. If the slope of the overpass AC follows a 3:4 ratio, what is the length of the slope AC?\nChoices:\nA: 10m\nB: 8m\nC: 6m\nD: 6√3m

**Input Image:** **Grid Image:**

**Outputs:**

### Rationales: The problem involves calculating the hypotenuse (slope AC) of a right triangle where one side (the height AB) is given, and the ratio of the two legs (AB:BC) is provided. Using these details, the Pythagorean theorem will help find the length of the hypotenuse.
### Let's think step by step.
### Step 1: <interleave>9,16,23,30<interleave>Identify the values and ratios given. The height of the bridge (AB) is 6 units, with a slope ratio of 3:4, indicating AB:BC.
### Step 2:<interleave>2,9,16,23,26,30,33<interleave>Assign the given ratio to the triangle's sides: AB corresponds to 3 parts, and BC corresponds to 4 parts.
### Step 3:<interleave>23,26,30,33<interleave>Calculate the length of BC. Since AB is 6 units (which corresponds to 3 parts), then each part is 2 units. BC, having 4 parts, will be 4 * 2 = 8 units.
### Step 4:Use the Pythagorean theorem, $\( AC = \sqrt{AB^2 + BC^2} \)$.
### Step 5:Substitute the values into the equation: $\( AC = \sqrt{6^2 + 8^2} = \sqrt{36 + 64} = \sqrt{100} = 10 \)$.
### The final answer is: A

Figure 6: An example from MINT-CoT dataset.

**Projector Ablation** We conduct an ablation study on the post interleave projector $P_{\text{post\_intlv}}$ and the post visual projector $P_{\text{post\_vis}}$ on the Interleaved CoT SFT stage. Both projectors were initially implemented as single-layer linear layers. We first remove both projectors entirely, and then replace them with two-layer MLPs using GELU activation. Both configurations are trained for three epochs. The results on the mathematical subset of MathVista are shown in Table 7, in which we find that the initial configuration as single-layer linear layers performs the best over all primary tasks.

## A.6 Additional Qualitative Results

In addition to Section 3.4, we provide more qualitative results of the baseline model Qwen2-VL-7B-Instruct and our proposed model MINT-CoT-7B in Figures 9 to 11.

## A.7 Limitations

In our dataset curation pipeline, the use of GPT-4o for dataset annotation still has some computational cost. Moreover, alternative reinforcement learning strategies beyond our method now remain underexplored.

**Question:**

Find the measure of the altitude drawn to the hypotenuse.
Choices:  A: 11  B: \\sqrt{297}  C: 27  D: 297

**Input Image:** **Grid Image:**

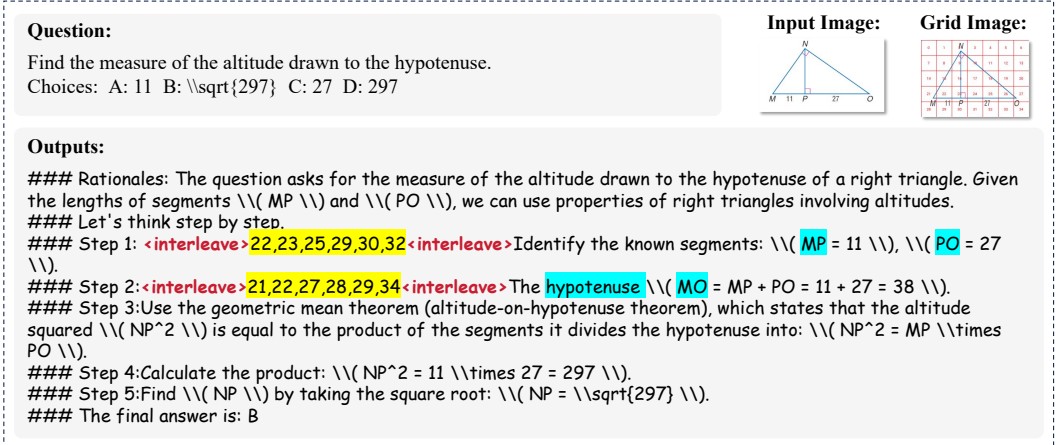

**Outputs:**

### Rationales: The question asks for the measure of the altitude drawn to the hypotenuse of a right triangle. Given the lengths of segments \\( MP \\) and \\( PO \\), we can use properties of right triangles involving altitudes.
### Let's think step by step.
### Step 1: <interleave>22,23,25,29,30,32<interleave>Identify the known segments: \\( MP = 11 \\), \\( PO = 27 \\).
### Step 2:<interleave>21,22,27,28,29,34<interleave>The hypotenuse \\( MO = MP + PO = 11 + 27 = 38 \\).
### Step 3:Use the geometric mean theorem (altitude-on-hypotenuse theorem), which states that the altitude squared \\( NP^2 \\) is equal to the product of the segments it divides the hypotenuse into: \\( NP^2 = MP \\times PO \\).
### Step 4:Calculate the product: \\( NP^2 = 11 \\times 27 = 297 \\).
### Step 5:Find \\( NP \\) by taking the square root: \\( NP = \\sqrt{297} \\).
### The final answer is: B

Figure 7: An example from MINT-CoT dataset.

**Question:**

Determine the degree measure of angle A in the figure, where circle O is inscribed in right triangle ABC. Angle ACB measures 90 degrees and angle CBO measures 30 degrees.
Choices:  A: 15°  B: 30°  C: 45°  D: 60°

**Input Image:** **Grid Image:**

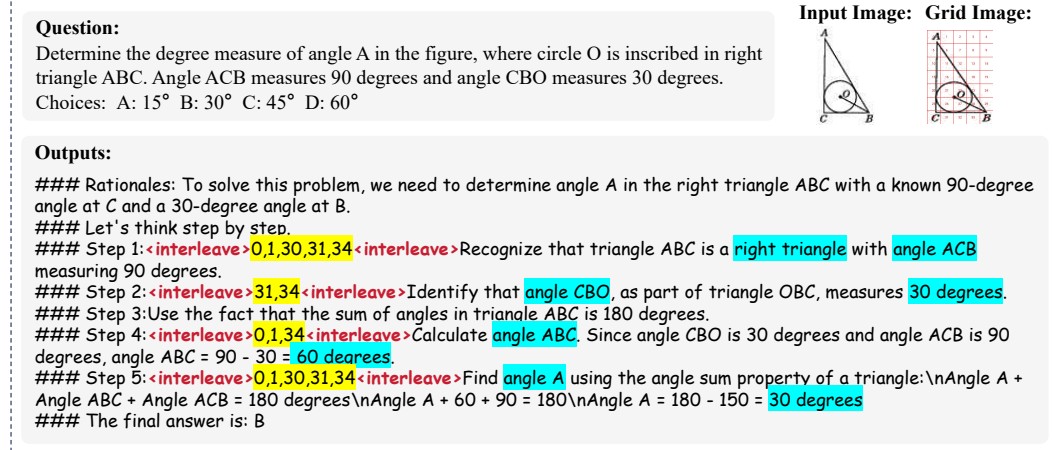

**Outputs:**

### Rationales: To solve this problem, we need to determine angle A in the right triangle ABC with a known 90-degree angle at C and a 30-degree angle at B.
### Let's think step by step.
### Step 1:<interleave>0,1,30,31,34<interleave>Recognize that triangle ABC is a right triangle with angle ACB measuring 90 degrees.
### Step 2:<interleave>31,34<interleave>Identify that angle CBO, as part of triangle OBC, measures 30 degrees.
### Step 3:Use the fact that the sum of angles in triangle ABC is 180 degrees.
### Step 4:<interleave>0,1,34<interleave>Calculate angle ABC. Since angle CBO is 30 degrees and angle ACB is 90 degrees, angle ABC = 90 - 30 = 60 degrees.
### Step 5:<interleave>0,1,30,31,34<interleave>Find angle A using the angle sum property of a triangle:\nAngle A + Angle ABC + Angle ACB = 180 degrees\nAngle A + 60 + 90 = 180\nAngle A = 180 - 150 = 30 degrees
### The final answer is: B

Figure 8: An example from MINT-CoT dataset.

**Question:**

As shown in the figure, in diamond ABCD, the diagonals AC and BD intersect at O. It is known that BD = 6 and AC = 8. The perimeter of diamond ABCD is ().

Choices: A: 40  B: 20  C: 10  D: 5

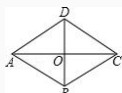

**Qwen2vl-7B-Instruct**

The quadrilateral ABCD is a rhombus since its diagonals are perpendicular. \n\n[asy]\npair A,B,C,D,O;\n\nA = (0,4);\nB = (6,0);\nC = (-3*sqrt(3), 3);\nD = (0, 4);\nO = (0, 0);\ndraw(A--B--C--D--cycle, black+linewidth(1));\ndraw(A--C, black+linewidth(1));\ndraw(B--D, black+linewidth(1));\ndraw(O--A, black+linewidth(1));\ndraw(O--B, black+linewidth(1));\ndraw(O--C, black+linewidth(1));\ndraw(O--D, black+linewidth(1));\nlabel("$A$", A, N);\nlabel("$B$", B, E);\nlabel("$C$", C, dir(315));\nlabel("$D$", D, N);\nlabel("$O$", O, S);\n[/asy]\n\nLet\'s label the intersection point of the diagonals O. \n\nSince OA = OC = 4 and OB = OD = 3, we have OB = 3 and BD = 6.\n\nSo, the perimeter of the rhombus ABCD is 4OA + 4OB = 4 * 4 + 4 * 3 = 16 + 12 = 28.\n\nTherefore, the answer is 28. \n\nSince the question asks for the perimeter of the diamond ABCD, the answer is 20. \n\nTherefore, the answer is B.   ✗

**MINT-CoT-7B**

### Rationales:

To find the perimeter of the rhombus, the length of all sides needs to be determined. Since the diagonals bisect each other at right angles in a rhombus, the sides can be found using the Pythagorean theorem.\n\n### Let's think step by step.

### Step 1:

Cosine Similarity Scores

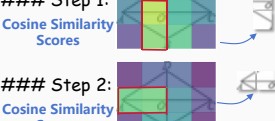

Identify the properties of the rhombus. In a rhombus, the diagonals bisect each other at right angles. Therefore, AO = OC = AC/2 = 8/2 = 4 and BO = OD = BD/2 = 6/2 = 3.

### Step 2:

Cosine Similarity Scores

Use the Pythagorean theorem to calculate the side length (AD) of the rhombus. The hypotenuse is AD, with AO = 4 and BO = 3.\n\n\\[\nAD = \\sqrt{AO^2 + BO^2} = \\sqrt{4^2 + 3^2} = \\sqrt{16 + 9} = \\sqrt{25} = 5\n\\]

### Step 3: Calculate the perimeter of the rhombus. Since all sides are equal in a rhombus, the perimeter is 4 times the side length.\n\nPerimeter \\( = 4 \\times 5 = 20 \\)

### The final answer is: B  ✓

Figure 9: Comparison between Qwen2-VL-7B-Instruct and MINT-CoT-7B.

---

**Question:**

As shown in the figure, the radius of circle O is 4.0, BC is the diameter, AC is the tangent of circle O, and AC = 6.0, then AB = ()

Choices:  A: 4  B: 6  C: 10  D: 12

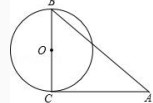

**Qwen2vl-7B-Instruct**

Because AC is tangent to circle O, it is perpendicular to OC. By defining the length of AC as x, we can solve for segment AB using the Pythagorean theorem in a right triangle:\n\n1. Start by understanding the Pythagorean theorem.\n2. Apply the Pythagorean theorem to a right triangle with segments AC and AB as the legs and segment BC as the hypotenuse.\n\nGiven that AO is the radius (4.0), we use the Pythagorean theorem in triangle OAB:\n\nAB = √(OA² + OB²) = √(4.0² + 4.0²) \nAB = √(16 + 16) \nAB = √32 \nAB ≈ 6.4\nSo the answer is approximate 6.4. It seems there is no exact option with this value, but the closest one would be C) 6. However, given the calculation, the length AB should be a bit more than the radius of circle O.\n\nTherefore, the answer is C:6.  ✗

**MINT-CoT-7B**

### Rationales: To determine AB, understand the properties of the circle and the right-angled triangle formed by the tangent AC and the diameter BC. Use the Pythagorean theorem in triangle ABC.

### Let's think step by step.

### Step 1:

Cosine Similarity Scores

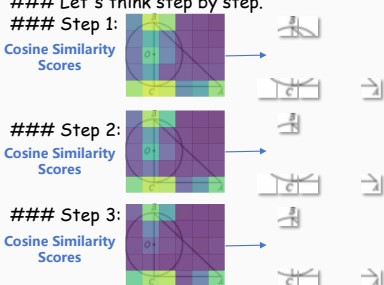

Identify the known lengths: radius of the circle (OC) is 4, and AC, the tangent, is 6.

### Step 2:

Cosine Similarity Scores

Recognize that BC is a diameter of the circle. Since the radius is 4, BC (being the diameter) is twice the radius: BC = 8.

### Step 3:

Cosine Similarity Scores

Since AC is a tangent and BC is a diameter, consider triangle ABC. Triangle ABC is a right triangle with AC perpendicular to BC.

### Step 4: Apply the Pythagorean theorem in triangle ABC: \\( AB^2 = AC^2 + BC^2 \\).
### Step 5: Substitute the known values into the Pythagorean theorem: \n\\( AB^2 = 6^2 + 8^2 \\).
### Step 6: Calculate the squares: \n\\( 6^2 = 36 \\) and \\( 8^2 = 64 \\).
### Step 7: Sum the squares: \n\\( AB^2 = 36 + 64 = 100 \\).
### Step 8: Take the square root to find AB: \n\\( AB = \\sqrt{100} = 10 \\).
### The final answer is: C  ✓

Figure 10: Comparison between Qwen2-VL-7B-Instruct and MINT-CoT-7B.

**Question:**

In ABCD, given that AB < BC, the perpendicular bisector of diagonal AC intersects AD at point E. Connect CE. If the perimeter of ABCD is 200cm, what is the perimeter of triangle CDE?

Choices:  A: 200.0  B: 400.0  C: 150.0  D: 100.0

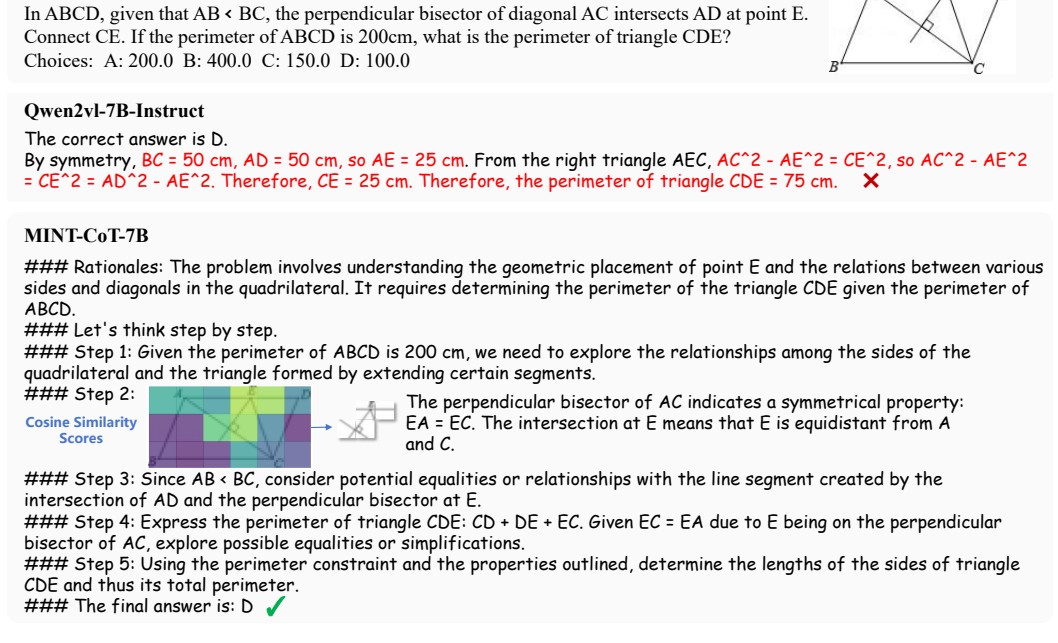

**Qwen2vl-7B-Instruct**

The correct answer is D.

By symmetry, BC = 50 cm, AD = 50 cm, so AE = 25 cm. From the right triangle AEC, AC^2 - AE^2 = CE^2, so AC^2 - AE^2 = CE^2 = AD^2 - AE^2. Therefore, CE = 25 cm. Therefore, the perimeter of triangle CDE = 75 cm.  ✗

**MINT-CoT-7B**

### Rationales: The problem involves understanding the geometric placement of point E and the relations between various sides and diagonals in the quadrilateral. It requires determining the perimeter of the triangle CDE given the perimeter of ABCD.

### Let's think step by step.

### Step 1: Given the perimeter of ABCD is 200 cm, we need to explore the relationships among the sides of the quadrilateral and the triangle formed by extending certain segments.

### Step 2:

**Cosine Similarity Scores**

The perpendicular bisector of AC indicates a symmetrical property: EA = EC. The intersection at E means that E is equidistant from A and C.

### Step 3: Since AB < BC, consider potential equalities or relationships with the line segment created by the intersection of AD and the perpendicular bisector at E.

### Step 4: Express the perimeter of triangle CDE: CD + DE + EC. Given EC = EA due to E being on the perpendicular bisector of AC, explore possible equalities or simplifications.

### Step 5: Using the perimeter constraint and the properties outlined, determine the lengths of the sides of triangle CDE and thus its total perimeter.

### The final answer is: D ✓

Figure 11: Comparison between Qwen2-VL-7B-Instruct and MINT-CoT-7B.

