# OpenReview forum: "MINT-CoT: Enabling Interleaved Visual Tokens in Mathematical Chain-of-Thought Reasoning"
_NeurIPS.cc/2025/Conference — NeurIPS 2025 poster_

### Official Review · Reviewer_XQF3 · 2025-06-09

**Clarity:** 3
**Significance:** 3
**Originality:** 3
**Rating:** 4
**Confidence:** 4

**Summary:**

This paper proposes a chain-of-thought method, MINT-CoT, for multimodal mathematical reasoning. Its core contribution lies in introducing the Interleave Token mechanism, which dynamically selects visual tokens related to the current mathematical semantics at each reasoning step and interleaves them into the textual reasoning process to achieve collaborative visual-textual reasoning. To support this method, the authors construct a 54K-scale MINT-CoT dataset and design a three-stage training process, gradually transitioning from purely textual supervision to reinforcement learning-based visual interleaving training.

**Questions:**

1.The paper does not specify whether there is any limit or sparsity constraint on the number of visual tokens inserted. Have there been instances where excessive tokens caused reasoning failures?
2.Does token selection behavior differ statistically between algebraic and geometric problems? Are there task-specific biases in token filtering?
3.You use GPT-4o to extract keywords in dataset annotation. Have you evaluated the impact of alternative extraction methods on final performance? There appears to be full reliance on GPT-4o, raising concerns about potential error propagation.
4.If OCR or keyword extraction fails, would visual token selection break down entirely? Have you incorporated such simulated error scenarios during training to improve robustness?

**Ethical Concerns:**

["NO or VERY MINOR ethics concerns only"]

**Final Justification:**

The authors have addressed my concerns.

**Limitations:**

No. The checklist claims that the limitations of the work are included in the appendix, but in fact, no such discussion could be found.

**Quality:**

3

**Strengths And Weaknesses:**

Strengths:
	1.The proposed Interleave Token mechanism enables dynamic insertion of visual tokens at each step of textual reasoning.
	2.The performance gains are significant, with improvements of 26.92% and 32.59% on GeoQA and MathVista-Math subtasks, respectively.

Weaknesses:
	1.The method essentially relies on traditional similarity matching principles; the Interleave Token is effectively a formalized expression of a query-like vector in an attention mechanism, lacking structural novelty.
	2.The method heavily depends on high-quality alignment between visual regions and reasoning steps in the dataset. Although the token index annotation process is automated, inaccuracies in the mapping can significantly degrade performance, limiting generalizability in real-world applications.
	3.Figure 3 is not clearly presented, with small fonts and repetitive data processing descriptions, making the methodological pipeline appear redundant and less intuitive. The coordination between textual explanation and illustration is suboptimal.
	4.Although the reinforcement learning stage adopts the GRPO strategy, there is no description of implementation details, making it difficult to assess the effectiveness of this phase.

---

> ### Author Rebuttal · Authors · 2025-07-31
>
> Thank you very much for your detailed feedback and constructive suggestions. We greatly appreciate your recognition of the Interleave Token mechanism and the strong performance gains achieved by MINT-CoT. Below, we address your concerns point by point.
> ### 1. Interleave Token Structure:
> > the Interleave Token is effectively a formalized expression of a query-like vector in an attention mechanism, lacking structural novelty.
>
> We would like to reclaim the novelty of our Interleave Token.  While the Interleave Token superficially resembles a query vector, its **design and function are distinct in several important aspects**:
>  - The Interleave token is a special token that is specifically designed for selecting visual tokens. It will be appended to the input sequences, which are passed forward through the entire transformer. And then it will compute the similarity between text content and visual tokens.
>  - In a highly integrated MLLM, the Interleave Token serves as a bridge that connects high-level CoT reasoning with low-level visual structure.
>  - Moreover, the Interleaved token is supervised during training, which is different from the query-like vector in the attention mechanism.
>
> ### 2. Mapping accuracy of visual regions in the dataset
> We agree that accurate alignment is important. To ensure annotation quality, we have designed our dataset curation pipeline with careful consideration and multiple safeguards:
>  - We used **state-of-the-art OCR tools** (e.g., PaddleOCR) to extract fine-grained elements such as symbols and labels.
>  - We employed GPT-4o, one of the strongest available MLLMs, to generate alignment annotations using specially designed prompts aimed at minimizing noise and ambiguity.
>
> Moreover, to validate the reliability of the automatic annotations, we conducted a **human evaluation**: 10 annotators independently reviewed 100 randomly sampled MINT-CoT examples. They assessed whether the selected visual tokens at each reasoning step were relevant and accurate. The results showed an average step-wise accuracy of **89.49%**, which demonstrates that our pipeline produces high-quality alignments suitable for training multimodal reasoning models.
>
> ### 3. Problem of Figure 3
> > with small fonts and repetitive data processing descriptions, making the methodological pipeline appear redundant and less intuitive. The coordination between textual explanation and illustration is suboptimal.
>
> Thanks for your suggestions. Given the complexity of the pipeline, we opted for a compact layout to keep it within a single figure. We encourage readers to zoom in for better readability. And we will refine it in the final version.
>
> ### 4. Inplemental details of GRPO strategy
> Due to space limitations in the main paper, we have provided the detailed implementation of the GRPO strategy in Sections A.4 and A.5 of the supplementary material. We will integrate these descriptions into the main text in the final version to improve clarity.
>
> ### 5. Limitation or sparsity constraint on the number of visual tokens inserted?
> After training, the number of interleaved visual tokens does not pose a risk of inference failure, as the selected tokens generally contain minimal irrelevant information. Furthermore, as presented in Table 1 in the supplementary material, the average number of Interleave Tokens per interleaved data point is 2.80, and the average number of selected indices per interleaved data point is 19.91, which reflects a sparse and controlled selection process, rather than excessive inclusion.
>
> ### 6. Behavior difference between algebraic and geometric problems
> > Does token selection behavior differ statistically between algebraic and geometric problems? Are there task-specific biases in token filtering?
>
> We analyzed whether token selection behavior differs between algebraic and geometric problems, using evaluation results of MINT-CoT-7B on MathVista-Math. Specifically, we compared samples from two distinct sources:
>  - UniGeo (geometry questions)
>  - FunctionQA (algebra/function questions)
>
> We found that, on average, the proportion of visual tokens selected during inference is 3.60% for geometry questions, while for algebra/function questions, the rate is 2.48%. This suggests that the model tends to select broader visual regions for geometry problems, reflecting a task-specific preference in token filtering.
>
> ### 7. GPT-4o annotation accuracy
> As discussed in **”2. Mapping accuracy of visual regions in the dataset”**, we conduct a human evaluation on the accuracy of our annotated data generated by GPT-4o. The results show an average per-step accuracy of 89.49%, indicating that the annotation pipeline is reliable and sufficiently precise for the task.
>
> ### 8. If OCR or keyword extraction fails, would visual token selection break down entirely?
> Overall, the annotation process handles OCR and keyword extraction failures well, and rarely leads to incorrect visual-token selection.
>  - Modern OCR tools (e.g., PaddleOCR) are highly accurate, and in our pipeline, OCR serves only as auxiliary input. If there are no OCR results in a image, the GPT-4o-based annotation process remains unaffected, as it does not depend on OCR outputs exclusively.
>  - During keyword extraction, prompts are carefully crafted to ensure that extracted keywords are both (1) highly related to the visual content, and (2) critical for reasoning. If no keyword satisfies these two conditions, no visual tokens will be interleaved for that reasoning step. As shown in Table 1 (supplementary), approximately 2,000 samples in the dataset contain no Interleave Tokens, which reflects this conservative design.
>
> ### 9. Limitations
> > The checklist claims that the limitations of the work are included in the appendix, but in fact, no such discussion could be found.
>
> Thank you for pointing this out — we apologize for the omission. The limitations of our current work are as follows: In the dataset curation pipeline, the use of GPT-4o for dataset annotation still has some computational cost. Moreover, alternative reinforcement learning strategies beyond our method now remain underexplored.
> We will include a discussion of these limitations in the final version of the paper.

---

> > ### Author Response · Authors · 2025-08-06
> >
> > Dear Reviewer XQF3,
> >
> > We are truly grateful for the time and effort you put into reviewing our work. To address your thoughtful concerns regarding our method, the reliability of the MINT-CoT dataset, and implementation clarity, we have provided a detailed rebuttal clarifying the unique function of the Interleave Token, the safeguards in our data pipeline, and the specifics of our training strategy.
> >
> > Therefore, we hope these responses have adequately addressed your comments, and we remain available for any further discussion. If our rebuttal has resolved your concerns, we would greatly appreciate your consideration of a higher rating, as it would be a strong encouragement for our work.
> >
> > Paper 784 authors

---

> > ### Comment · Reviewer_XQF3 · 2025-08-09
> >
> > Dear authors, you have addressed my concerns, and I have therefore decided to raise my score.

---

> > > ### Author Response · Authors · 2025-08-09
> > >
> > > Thank you very much for acknowledging our rebuttal and efforts!

---

### Official Review · Reviewer_nxuS · 2025-06-30

**Clarity:** 2
**Significance:** 3
**Originality:** 3
**Rating:** 4
**Confidence:** 5

**Summary:**

This paper proposes MINT-CoT, a novel approach to enhancing mathematical reasoning in multi-modal large language models (MLLMs) through interleaved token-level interleaving Chain-of-Thought (CoT). Central to this approach is the MINT-CoT dataset of 54K problems, where each reasoning step is explicitly grounded to relevant visual regions. The authors design a three-stage training pipeline involving text-only CoT supervised fine-tuning (SFT), interleaved CoT SFT, and interleaved CoT reinforcement learning (RL), culminating in the MINT-CoT-7B model. The model achieves strong gains on visual math benchmarks.

**Questions:**

1. Resolution Dependency of Token Selection: Since the number of visual tokens depends on input image resolution, how do you handle ground-truth labels for active visual tokens when resolution changes? Do you apply any post-processing to align token indices across different resolutions?
2. Efficiency Concerns: How many visual tokens are input during reasoning on average? Could you provide details about the training time and inference time overhead introduced by interleaving visual tokens compared to the baseline?
3. Number of Interleave Tokens: How many Interleave Tokens are leant during training?
4. Design Choice in Token Selection Loss (Eq. 7): Why is the token selection supervised with a binary cross-entropy loss (Eq. 7) rather than integrated directly into the next-token prediction objective (Eq. 6)? For instance, could token selection be formulated as a number secretion over L visual tokens, jointly optimized during sequence modeling? You can number the visual tokens from 1 to L, and predict which token number is selected.
5. Domain Specificity of Visual Tokens: What justifies the use of selected visual tokens specifically for geometric math reasoning? I think this token election can be generalize to natural images or more general domains.

**Ethical Concerns:**

["NO or VERY MINOR ethics concerns only"]

**Final Justification:**

**Thanks authors for providing detailed explanations and additional experiments.** I appreciate the contribution of proposing a pipeline that interleaves visual and textual tokens to support reasoning.

However, the effectiveness of this pipeline fundamentally relies on the accuracy of the selected visual tokens, which in turn depends on the perception ability of the visual encoder. Perception ability refers to the encoder’s capacity to extract, represent, and highlight key features from raw diagrams, an essential prerequisite for any visual reasoning task. But throughout the rebuttal, the authors repeatedly emphasis that current MLLMs exhibit limited visual perception capabilities and that they do not attempt to enhance the encoder's perception performance. This raises a critical concern: without accurate visual perception, how can one ensure that the interleaved Chain-of-Thought (CoT) reasoning steps are grounded in correct visual understanding?
If the visual inputs are poorly encoded, the downstream reasoning, even if structurally improved,  still operate on incorrect or incomplete visual representations.

Based on the authors’ experiments, in the setting where the model is trained without inserted visual tokens, but with visual supervision loss (“Reasoning w.o. Visual Tokens”), the performance drops even below the baseline (Text-only CoT SFT). This result raises a serious concern: the proposed training data may not provide meaningful learning signals for the visual encoder. The quality of the visual token labeling used for supervision has also been questioned by two other reviewers.

Overall, without ensuring accurate visual token selection, the performance improvement observed from interleaving with textual tokens still lacks a clear and convincing explanation. Without deeper insights into the underlying mechanism behind these gains, the proposed pipeline of interleaving visual and textual tokens for reasoning currently appears more like an engineering workaround or technical report, rather than a fully substantiated scientific contribution suitable for a top-tier venue NeurIPS.
Regarding the questions about the resolution dependency of token selection/ Application on other visual reasoning tasks, I have asked this twice but have yet to receive a clear answer from the authors.

**I keep my score, but lean more toward the borderline.**

**Limitations:**

Yes.

**Paper Formatting Concerns:**

I have no formatting concerns.

**Quality:**

2

**Strengths And Weaknesses:**

## Strengths:
1. The paper addresses a fundamental challenge in multi-modal mathematical reasoning.
2. The introduction of an Interleave Token that dynamically selects arbitrary-shaped visual regions per reasoning step is novel.
3. The construction of a large-scale dataset (54K samples) with token-level alignment between text and vision is a significant contribution to the community.
4.  MINT-CoT-7B achieves substantial performance improvements over baselines on two challenging benchmarks (MathVista and GeoQA).

## Weaknesses:
1. Unclear Visual Perception Capability: While the paper emphasizes fine-grained interleaving of visual tokens, it relies on a math-domained variant of CLIP (MAVIS) for visual grounding. However, CLIP and its derivatives are known to capture coarse-grained semantics, which raises concerns about their ability to accurately perceive fine-grained visual elements such as small geometric cues, labels, or spatial relationships crucial in mathematical diagrams. The paper does not provide direct evidence that MINT-CoT meaningfully overcomes these perceptual limitations.  Although the model shows improved final-answer accuracy on benchmarks like MathVista and GeoQA, it remains unclear whether the interleaved visual steps are always semantically or visually correct. There is no diagnostic analysis or qualitative study examining failures in intermediate reasoning, which could be masked by correct final answers but reflect flaws in the model’s step-wise visual understanding.
2. Questionable Necessity of Multimodal CoT for Geometric Figures:  The paper assumes that visual interleaving is beneficial for geometric figures. However, many such diagrams consist of simple symbolic elements—lines, points, circles, triangles—which lack rich semantics and can be easily and precisely described in natural language. Given that LLMs are already capable of understanding these concepts from large-scale text-only training, it's unclear whether introducing visual tokens truly adds value. In fact, there is no direct evidence provided to show that the model understands these visual tokens better than their natural language descriptions. For example, does the model truly understand the meaning of the selected visual tokens? what happens if a portion of these selected tokens is randomly dropped or replaced with irrelevant ones?
3. Unclear Advantage of Token-Level Visual Grounding over Box-Level: While the paper criticizes box-level grounding for including irrelevant visual content, visual tokens are also subject to noise and overlap, especially in complex diagrams. For example, a token covering a "circle" may still partially include other elements (e.g., lines or text). Furthermore, the empirical improvement in Table 4 over the box-based OCT baseline is minimal.
4. Missing Ablation on the Effectiveness of Interleaved CoT: The paper lacks a key ablation study to isolate the impact of the proposed interleaved multimodal CoT reasoning. Specifically, it remains unclear how much of the performance gain comes from the MINT-CoT dataset itself versus the interleaving strategy. A natural baseline would be to train the model on the MINT-CoT dataset without interleaved visual tokens, using only text-based CoT supervision, and compare performance. Without this, it is difficult to determine whether the interleaved design is truly responsible for the observed improvements or whether the gains stem largely from exposure to large-scale curated training data.

---

> ### Author Rebuttal · Authors · 2025-07-31
>
> Thank you very much for your feedback and positive comments. You highly praised the design of our Interleave Token, our large-scale dataset, and the performance of MINT-CoT-7B. Your concerns are addressed below.
>
> ### 1. Visual Perception Capability
> > it relies on a math-domained variant of CLIP (MAVIS) for visual grounding …
>
> We would like to clarify that **our MINT-CoT framework does not rely on MAVIS or any math-domain variant of CLIP**. We apply our MINT-CoT framework on Qwen2-VL-7B-Instruct model. As stated in lines 46-53, models like CLIP and SigLIP are pre-trained on natural images with general scenes, making mathematical images out-of-distribution. So attention-based token selection method to identify relevant visual tokens during mathematical reasoning is difficult.
>
> > The paper does not provide direct evidence that MINT-CoT meaningfully overcomes these perceptual limitations …
>
> We do not aim to improve the raw perception ability of the encoder, but rather provide an **alternative mechanism** through the interleaving of selected visual tokens at each reasoning step, which helps the model to “watch” these important visual tokens again.
>
> ### 2. Necessity of Multimodal CoT for Geometric Figures
> #### 2.1. Disadvantages of text-only reasoning
> > Given that LLMs are already capable of understanding these concepts from large-scale text-only training …
>
> In practice, text-only captions of mathematical figures are frequently inaccurate. As shown in MAVIS, the visual perception of models like CLIP is inaccurate on mathematical reasoning tasks, so the natural language captions generated from MLLMs are often inaccurate. As a result, we need to interleave the related visual regions during reasoning, which can significantly reduce hallucination and improve the accuracy of mathematical reasoning.
>
> #### 2.2. A training-free experiment
> > In fact, there is no direct evidence provided to show that the model understands these visual tokens better than their natural language descriptions …
> We have conducted a **training-free** multimodal CoT experiment on our base model:
>  - Input question: *As shown in the figure, AB is tangent to circle O at point B, if angle A = 36.0, then angle C is equal to ()\nChoices:\nA:36°\nB:54°\nC:60°\nD:27°*
>  - Output solution: *Since AB is tangent to circle O at point B, we have angle ABO = 90°. Therefore, angle AOB = 90° - angle A = 54°. Since OA = OB, we have angle OAB = angle OBA = 36°. Therefore, angle C = 180° - angle AOB - angle OAB = 180° - 54° - 36° = 90°. Since angle C is an inscribed angle, we have angle C = 1/2 angle AOB = 1/2 * 54° = 27°. Therefore, the answer is D.*
>
> The output solution of the base model is wrong when it comes to “Since OA = OB, we have angle OAB = angle OBA = 36°”. So we manually concat the related visual tokens before this reasoning step, and find that the whole output solution will be:
>  -  *Since AB is tangent to circle O at point B, we have angle ABO = 90°. Therefore, angle AOB = 90° - angle A = 54°.<selected_visual_tokens> Therefore, angle C = 1/2 angle AOB = 27°. Therefore, the answer is D.*
> So, the model outputs with correct answer when interleaving correct visual tokens during reasoning, which can proves that MLLMs benefits from Multimodal CoT when solving methematical problems.
>
> #### 2.3. Randomly select visual tokens
> > what happens if a portion of these selected tokens is randomly dropped or replaced with irrelevant ones?
> We conduct an experiment that randomly selects visual tokens during inference of the Interleave CoT SFT model. The results show that inference with random selections introduced irrelevant visual context in each reasoning step, which deeply hurts the original format and logic.
> | Qwen2-VL-7B-Instruct | Interleave CoT SFT | Interleave CoT SFT (random select) |
> | - | - | - |
> | 41.11 | 67.78 | 31.11 |
>
>
> ### 3. Precision of token selection
> > visual tokens are also subject to noise and overlap…
>
> Due to the limitation of the patching method of ViT, which is not fine-grained enough, it’s hard to avoid some noise when selecting visual tokens. However, token-level grounding remains significantly more precise than box-level selection.
> > Furthermore, the empirical improvement in Table 4 over the box-based OCT baseline is minimal.
>
> As shown in Table 4 in our paper, the Bounding Box CoT SFT is 2.22% lower than Interleaved CoT SFT on MathVista-Math, and even performs worse than Text-only CoT SFT on GEO and GPS domains, which demonstrates that the Bounding Box CoT SFT is worse than our Interleaved CoT SFT method.
>
> ### 4. Text-only CoT Baseline
> Thanks for your suggestion. We conduct experiments on the text-only SFT followed by text-only RL. As shown below, MINT-CoT still outperforms this stronger baseline:
> | Model | Qwen2-VL-7B-Instruct | Text-only CoT SFT | Interleaved CoT SFT |  Text only SFT + Text only RL | MINT-CoT-7B |
> | - | - | - | - | - | - |
> | MathVista-Math | 41.11    | 64.07                | 67.78               | 70.74 | **73.70** |
>
> This confirms that our improvements specifically arise from the interleaved visual reasoning enabled by MINT-CoT.
>
> ### 5. Resolution Dependency of Token Selection
> When gridding the image, we align our gridding method and image resolution with the Qwen image processing method, so the selected token indices are correct during training. For models with different patching strategies, we support token index remapping based on image resize ratios, so the selected visual tokens remain aligned.
>
> ### 6. Efficiency Concerns
> > How many visual tokens are input during reasoning on average?
>
>  - As presented in Table 1 in the supplementary material, the Average number of Interleave Tokens per interleaved data point is 2.80, and the Average number of selected indices per interleaved data point is 19.91.
>  - Importantly, when interleaving the selected visual tokens, we just concat the visual embeddings to the text embeddings, rather than re-encoding the image through the vision encoder (ViT). This design reduces a lot of time.
>  - Training time and Inference time:
> We provide the training time of three training stages on 8*A100 GPUs, and the average inference time per question on GeoQA:
> |  | Text-only CoT SFT | Interleave CoT SFT | Interleave CoT RL |
> | - | - | - | - |
> | Training | 4h | 9h | 25h |
> | Inference | 8.15s | 8.03s | 8.24s |
>
>
> ### 7. Number of Interleave Tokens
> > How many Interleave Tokens are leant during training?
>
>
> As presented in Table 1 in the supplementary material, the average number of Interleave Tokens per interleaved data point is 2.80.
>
> ### 8. Design Choice in Token Selection Loss
> Thank you for the insightful suggestion. But we concern that it will not be efficient enough to select visual tokens through next-token prediction. Moreover, the order of selected visual tokens is not semantically meaningful, and attempting to predict them autoregressively may introduce unnecessary complexity and training instability. We wil explore this formulation in the future work.
>
> ### 9. Application on other visual reasoning tasks:
> > What justifies the use of selected visual tokens specifically for geometric math reasoning? I think this token election can be generalize to natural images or more general domains.
>
> Thanks for your insight. While the current focus of our research is specifically on multi-modal math problems, our method can also be applied to other visual reasoning tasks.
>  - Our dataset curation pipeline can be adapted to other tasks. The image gridding strategy, auxiliary OCR tools, keyword extraction, and final visual index annotation can also be applied to general visual reasoning tasks. For math reasoning tasks, we need to align fine-grained visual indices with the reasoning steps. In contrast, general visual reasoning tasks do not require such a high level of granularity, so our pipeline remains suitable for general tasks as well.
>  - The introduction of an Interleave Token for grounding reasoning steps in corresponding visual content is a general mechanism that enhances multimodal alignment and interpretability across various tasks.
>  - Our three-stage training pipeline can be easily adapted to many other general tasks.

---

> > ### Comment · Reviewer_nxuS · 2025-08-03
> >
> > **Thank authors for providing the rebuttal. Some of my concerns were partially addressed, but a few responses do not fully or faithfully answer my questions, and additional concerns have arisen based on the responses provided.**
> >
> > 1. From authors: We do not aim to improve the raw perception ability of the encoder, but rather provide an alternative mechanism through the interleaving of selected visual tokens at each reasoning step, which helps the model to “watch” these important visual tokens again.
> >
> > If the visual encoder itself lacks strong perception ability, how can it effectively help the model “watch” important visual tokens again? Additionally, how can you ensure that the interleaved visual cues are faithful and accurately reflect the relevant content?
> >
> > 2.  For the second question *Necessity of Multimodal CoT for Geometric Figures* , the authors did not carefully check my point. I mentioned that LLMs are already capable of understanding concepts from large-scale text-only training, but the authors answered with points like “text-only captions of mathematical figures are frequently inaccurate” and “the natural language captions generated from MLLMs are often inaccurate.” I did not mention anything about how MLLMs generate image captions.
> >
> > My point in this question is: how do you ensure that the model understands these abstracted visual tokens better than their natural language descriptions? For example, how do visual selection tokens for triangle ABC outperform textual tokens like “triangle ABC”? You also mentioned in your answer that the visual perception of models is inaccurate on mathematical reasoning tasks—so why would the selected visual tokens be better understood than textual tokens?
> >
> > For the case you provided, comparing your model responses to the base model does not sufficiently support the claimed observation, since your model was trained with additional math corpus data. A more controlled comparison would be to train models on the same scale of data, but replace the interleaved visual tokens with equivalent textual descriptions.
> > Moreover, authors could train the model using the token selection loss, but without interleaving visual tokens during reasoning. This would help disentangle how much of the improvement comes from additional visual supervision, versus truly understanding the meaning of visual tokens within CoT reasoning
> >
> > 3. I understand that during training, the interleaved tokens are supervised using the labels for selected visual tokens.
> >
> > However, during inference, it seems the model does not explicitly use the selected visual tokens extracted from visual feature maps. The authors did not provide a description of this detail, but I guess it uses placeholders or special tokens. I also noticed that training time roughly doubles, while inference time remains unchanged.
> >
> > Could the authors clarify why performance improves even though the interleaved tokens are not explicitly used during inference? What are the actual benefits introduced during training that lead to better reasoning, despite the apparent absence of interleaved visual features at test time?

---

> > ### Comment · Reviewer_nxuS · 2025-08-05
> >
> > Thank authors for appreciating the continued conversation with the reviewers, as commented in your response to Reviewer PFDq (willingness to engage further and your appreciation for the consideration of a higher rating.)
> >
> > After reviewing your first-round responses, I noticed that some of points require further clarification (I listed in my initial response) and that some questions remain unanswered.
> >
> > Unanswered question about **resolution dependency of token selection**: Your ground truth is constrained by the feature resolution. How do you handle the ground-truth labels for active visual tokens when the resolution changes?
> >
> > **Application on other visual reasoning tasks**: the authors respond that ''for math reasoning tasks, we need to align fine-grained visual indices with the reasoning steps. In contrast, general visual reasoning tasks do not require such a high level of granularity...''.  What type of “granularity” you are referred to? visual granularity or something else?
> >
> > If you pointed visual granularity. In fact, natural images often require more fine-grained visual cues than abstract diagrams. Natural images carry rich semantic information and often require recognition of fine details (pixel-level segmentation), e.g., identifying a dog may involve detecting parts like eyes, nose, ears, furs. Thus, how could you make sure ''so our pipeline remains suitable for general tasks as well..''

---

> > > ### Author Response · Authors · 2025-08-06
> > >
> > > Thank you for providing us the chance for communication. We address your concerns below:
> > > ### 1. Resolution dependency of token selection
> > > > How do you handle the ground-truth labels for active visual tokens when the resolution changes?
> > >
> > > We address resolution changes through a systematic process. When the resolution changes:
> > > 1. Since images are divided into patches of a fixed patch size (28 in qwen2vl), during the image processing period of training, Qwen2vl will first resize the image to the nearest resolution divisible by the patch size if the original dimensions are incompatible.
> > > 2. Next, we map the original ground-truth token selections to the new resolution. Our mapping ensures that the new ground-truth tokens fully cover the original selected regions to preserve critical information, while minimizing the number of tokens required.
> > >
> > > This approach guarantees consistency in token selection across varying resolutions during training and inference.
> > >
> > > ### 2. Application on other visual reasoning tasks
> > > > What type of “granularity” you are referred to?
> > >
> > > The term "granularity" here refers to the level of detail required for token selection. In mathematical reasoning tasks, images often contain structured graphs where relevant tokens may be irregularly distributed or highly discrete (e.g., individual line segments or scattered geometric elements), which are hard to ground. In contrast, general visual reasoning typically deals with object-centric regions that naturally conform to contiguous shapes like bounding boxes. For example, as you said, “identifying a dog may involve detecting parts like eyes, nose, ears, furs”, while these places can often be grounded by bounding boxes.
> > >
> > > While natural images may require pixel-level precision in some cases, our method's token selection mechanism is flexible enough to handle both:
> > > 1. Irregular distributions needed for mathematical diagrams
> > > 2. Contiguous regions (rectangular or otherwise) suitable for object-centric reasoning
> > >
> > > Thus, our method can adapt to broader visual reasoning tasks.

---

> > > ### Author Response · Authors · 2025-08-08
> > >
> > > Dear Reviewer nxuS,
> > >
> > > As the rebuttal period is drawing to a close, we would like to sincerely thank you for the time, effort, and detailed feedback you have provided throughout the process. Your questions have been very helpful in guiding us to strengthen our work.
> > >
> > > We hope that our latest clarifications and results have addressed your earlier concerns and resolved the doubts you raised. If there are still any remaining points that you feel require further evidence or explanation, please let us know, and we would be happy to provide additional details promptly.
> > >
> > > Thank you again for your engagement and constructive discussion.
> > >
> > > Paper 784 authors

---

> ### Author Response · Authors · 2025-08-05
>
> Thank you for your follow-up questions and for engaging deeply with our work. We appreciate the opportunity to clarify these important points. Your feedback is invaluable, and we hope our detailed responses below will fully address your remaining concerns.
> ### Question 1:
> > If the visual encoder itself lacks strong perception ability, how can it effectively help the model “watch” important visual tokens again?
>
> Sorry for making you confused. We would like to clarify that the lack of perception ability of visual encoders refers to that, the LLM backbone cannot well understand the visual embedding of the vision encoder on mathematical images. Since these images are out-of-distribution for the vision encoder, the features of significant mathematical regions, e.g., geometric characteristics, are not highlighted enough within their embeddings, which causes the LLM to fail to retrieve or associate the relevant visual tokens correctly while solving the problem. For example, when the reasoning step comes to “triangle ABC”, sometimes the LLM doesn’t know where “triangle ABC” is, which is “triangle ABC” in the image, or misreads it as “triangle BCD”.
>
> Instead, our design using the interleave token specifically targets retrieving these important visual tokens and interleaving them in CoT, which explicitly guides the LLM to attend to the correct region and mitigates hallucinations. Such a token selection process is not dependent on the unsatisfactory capabilities of the visual encoder, but rather on our supervised fine-tuning with MINT-CoT datasets.
>
> So we do not try to improve the perception ability, but to design a method that helps to provide the LLM with the related visual embeddings in a specific reasoning step.
>
> > Additionally, how can you ensure that the interleaved visual cues are faithful and accurately reflect the relevant content?
>
> We ensure the accuracy both through dataset curation and the training process.
> 1. To ensure annotation quality, we have designed our dataset curation pipeline with careful consideration and multiple safeguards:
>  - We used state-of-the-art OCR tools (e.g., PaddleOCR) to extract fine-grained elements such as symbols and labels.
>  - We employed GPT-4o, one of the strongest available MLLMs, to generate alignment annotations using specially designed prompts aimed at minimizing noise and ambiguity.
>
> Moreover, to validate the reliability of the automatic annotations, we conducted a **human evaluation**: 10 annotators independently reviewed 100 randomly sampled MINT-CoT examples. They assessed whether the selected visual tokens at each reasoning step were relevant and accurate. The results showed an average step-wise accuracy of **89.49%**, which demonstrates that our pipeline produces high-quality alignments suitable for training multimodal reasoning models.
>
> 2. We conduct Interleaved CoT SFT to help the model learn how to select correct visual tokens. As shown in Figure 4 in the paper, the F1 score exhibits an upward trend during training, which makes the interleaved tokens faithful and accurate as much as possible

---

> ### Author Response · Authors · 2025-08-05
>
> ### Question 2:
> > how do you ensure that the model understands these abstracted visual tokens better than their natural language descriptions? For example, how do visual selection tokens for triangle ABC outperform textual tokens like “triangle ABC”?
>
> Sorry for making you confused.  We think the natural language descriptions and our interleaved visual tokens are not supposed to compare in this task and setting, for the following two reasons:
>
> 1.  **MLLM cannot well understand the original visual embeddings**, so it's hard for it to generate accurate natural language descriptions during reasoning. The inaccurate math descriptions would harm the reasoning quality. Instead, our interleave token selection method overcomes it by selecting visual tokens using the similarity between the hidden states of the interleave token and visual tokens. This approach helps the model to locate important regions, but does not enforce it to describe them using language.
>
> 2. Further, these two settings (the natural language descriptions and our interleaved visual tokens) **are not comparable alternatives, but rather a sequential relationship**. Only when the model accurately locates the visual tokens (like what we do), can the model generate accurate descriptions. Therefore, our method is more like a foundation for accurate natural language descriptions because reasoning only with textual description is hard. This will be proved in our ablation study below.
>
> > You also mentioned in your answer that the visual perception of models is inaccurate on mathematical reasoning tasks—so why would the selected visual tokens be better understood than textual tokens?
>
> As we clarified in Question 1, the visual tokens can help the model to locate the visual regions where text-only inference mode has hallucinations. It is the structured interleaving that helps the model associate visual and textual modalities more reliably, which improves reasoning accuracy.
>
> > A more controlled comparison would be to train models on the same scale of data, but replace the interleaved visual tokens with equivalent textual descriptions.
>
> Thanks for your suggestion. Following your suggestion, we conducted an ablation study where we replaced the interleaved visual tokens with their corresponding textual keywords (e.g., "angle AOB", "triangle ABC") and add a prefix like: "This step will reason with …". In this modified setting, the model was trained using only textual descriptions of key visual regions rather than interleaved visual tokens.
>
> When trained with identical parameters to our Interleaved CoT SFT stage, the model demonstrated inferior performance to Text-only CoT SFT, which may be due to the redundancy of the keywords. It is also well below our Interleaved CoT SFT. This result confirms that accurate visual grounding through our token selection method is helpful for generating correct textual descriptions during reasoning.
>
> | Model | Qwen2-VL-7B-Instruct | Text-only CoT SFT | Interleaved CoT SFT |  Text-only visual keywords CoT SFT|
> | - | - | - | - | - |
> | MathVista-Math | 41.11    | 64.07                | 67.78               | 62.96 |
>
> A concrete failure case illustrates this point clearly: “... Since E is the midpoint of angle FEB, angle FEB = 2 times angle AEF = 40 circ …” In fact, E is the midpoint of **arc** FEB, rather than **angle** FEB, so angle FEB doesn’t eual to 2 times angle AEF. This wrong textual description leads the model to reason in the wrong direction. In contrast, our MINT-CoT method avoided this error, which shows that our method is more like a foundation for accurate natural language descriptions, helping the MLLM for better reasoning.
>
> > Moreover, authors could train the model using the token selection loss, but without interleaving visual tokens during reasoning.
>
> As you suggested, we conduct an experiment that trains with the interleave tokens, while without visual tokens inserted during reasoning.
>
> | Model | Qwen2-VL-7B-Instruct | Text-only CoT SFT | Interleaved CoT SFT | Reasoning w.o. Visual Tokens |
> | - | - | - | - | - |
> | MathVista-Math | 41.11    | 64.07                | 67.78               | 63.3 |
>
> As shown in the table, the performance drops when the interleaved visual tokens are removed during inference. This experiment demonstrates that the performance gain comes from the model truly understanding and utilizing the interleaved visual tokens within the CoT reasoning process, not just from the additional visual supervision during training.

---

> > ### Author Response · Authors · 2025-08-05
> >
> > ### Question 3:
> > > However, during inference, it seems the model does not explicitly use the selected visual tokens extracted from visual feature maps. The authors did not provide a description of this detail, but I guess it uses placeholders or special tokens.
> >
> > Sorry for the confusion caused. The model explicitly uses the selected visual tokens. Like we stated in paragraph “Inference with Interleaved Visual Tokens.” in Section 2.1: “​​With the selected visual tokens  obtained at each reasoning step, MINT-CoT interleaves both visual content and text-based reasoning steps throughout the inference process.” More specifically, we extract the visual embeddings of the visual tokens, and concatenate them to the textual embeddings during inference, thus forming the interleaved reasoning chain.
> >
> > > Could the authors clarify why performance improves even though the interleaved tokens are not explicitly used during inference? What are the actual benefits introduced during training that lead to better reasoning, despite the apparent absence of interleaved visual features at test time?
> >
> >  1. **Training time:** The training time doubles because the training epoch and steps are different. For text-only SFT, we only train for 2 epochs, while for interleaved cot SFT, we train for 3 epochs. So the comparison of the training time per epoch between the two stages is:
> >
> > |  | Text-only CoT SFT | Interleave CoT SFT |
> > | - | - | - |
> > | Training time | 4h | 9h |
> > | Training time per epoch | 2h | 3h |
> >
> >  2. **Inference Time:** The inference time remains nearly constant due to two counteracting effects:
> >  * **High Efficiency of Interleaving:** As stated in our original rebuttal, this process is very fast because we only concatenate cached embeddings. We do not re-run the vision encoder, avoiding significant computational overhead.
> >  * **Reduction in Failure Cases:** Our Interleaved CoT SFT model and MINT-CoT-7B model makes fewer mistakes. The less accurate baselines frequently fall into failure modes, such as generating nonsensical text until the maximum token limit is reached or engaging in lengthy and incorrect self-correction loops. The reduction in these time-consuming failure cases in our model effectively offsets the minor overhead of concatenating the visual embeddings.

---

### Official Review · Reviewer_PFDq · 2025-06-30

**Clarity:** 2
**Significance:** 2
**Originality:** 3
**Rating:** 3
**Confidence:** 3

**Summary:**

The paper presents a method for visual math reasoning where an interleave token at each reasoning step is used to select additional relevant visual tokens for reasoning.

**Questions:**

As listed in the "weaknesses" section above, copied below

- Need to clarify the use of "interleave tokens".  e.g., Note that an interleave token is the output of a reasoning step. (without visual tokens), Do we assume that the interleave token might not be optimal since it is generated without visual info, or do we simply assume that the ground-truth reasoning steps are given -- therefore the proposed method is a data curation method that enhance the reasoning steps with additional visual tokens?
- the method is task or dataset specific. I am not convinced that the method can be easily applied to other visual reasoning tasks.
- the evaluation is performed only on MathVista. More, different benchmarks need to be used to validate how general the method can be applied.

**Ethical Concerns:**

["NO or VERY MINOR ethics concerns only"]

**Limitations:**

yes

**Quality:**

2

**Strengths And Weaknesses:**

Strengths: the research is well-motivated. The proposed method is conceptually simple, and its effectiveness is validated on a standard visual math benchmark.

Weaknesses:
- the method is task or dataset specific. I am not convinced that the method can be easily applied to other visual reasoning tasks.
- the evaluation is performed only on MathVista. More, different benchmarks need to be used to validate how general the method can be applied.
- Need to clarify the use of "interleave tokens".  e.g., Note that an interleave token is the output of a reasoning step. (without visual tokens), Do we assume that the interleave token might not be optimal since it is generated without visual info, or do we simply assume that the ground-truth reasoning steps are given -- therefore the proposed method is a data curation method that enhance the reasoning steps with additional visual tokens?

---

> ### Author Rebuttal · Authors · 2025-07-31
>
> Thank you for your review and feedback. We sincerely appreciate your recognition of the motivation and simplicity of our approach. We address your concerns in detail below.
> ### 1. Application on other visual reasoning tasks:
> > The method is task or dataset specific. I am not convinced that the method can be easily applied to other visual reasoning tasks.
>
> While the current focus of our research is specifically on multi-modal math problems, our method can also be applied to other visual reasoning tasks.
>  - Our dataset curation pipeline can be adapted to other tasks. The image gridding strategy, auxiliary OCR tools, keyword extraction, and final visual index annotation can also be applied to general visual reasoning tasks. For math reasoning tasks, we need to align fine-grained visual indices with the reasoning steps. In contrast, general visual reasoning tasks do not require such a high level of granularity, so our pipeline remains suitable for general tasks as well.
>  - The introduction of an Interleave Token for grounding reasoning steps in corresponding visual content is a general mechanism that enhances multimodal alignment and interpretability across various tasks.
>  - Our three-stage training pipeline can be easily adapted to many other general tasks.
>
> ### 2. Evaluation on other benchmarks
> > More different benchmarks need to be used to validate how general the method can be applied.
>
> We appreciate this suggestion and have extended our evaluation to cover additional datasets. Specifically:
>  - In our paper, the results of **GeoQA** have been presented.
>  - In our supplementary materials, we have added the evaluation results on the Mathematics section of **MMStar** benchmark (Table 2 in Suppl.)
>  - Due to the time limit, we add evaluation experiments on the Vision Only domain of **Mathverse**, which can well represent the improvement of our method.
>  - Moreover, we add experiments on the Mathematics section of the **MMMU-Pro-V**. Although this subset contains only 60 questions, our method still achieved the best performance.
>
> The combined results of those benchmarks are shown below:
> | Benchmark  | Qwen2-VL-7B-Instruct  | Text-only CoT SFT | Interleaved CoT SFT | Interleaved CoT RL |
> |----------------------------------|--------------------|----------------------|---------------------|----------------|
> | MathVista-Math | 41.11              | 64.07                | 67.78               | **73.70**       |
> | GeoQA                            | 37.80              | 59.02                | 62.07               | **64.72**          |
> | MMStar-Math                      | 46.4               | 67.6                 | 68.0                | **69.6**           |
> | MathVerse_MINI (Vision Only)     | 23.98        | 32.74              | 33.25            | **36.42**    |
> | MMMU_Pro_V                       | 25                 | 25                   | **30**                  | **30**             |
>
>
> ### 3. Clarification of the use of "interleave tokens"
> > Note that an interleave token is the output of a reasoning step. (without visual tokens) …
>
> Sorry for the confusion caused. We need to clarify that Interleave Token is not *”the output of a reasoning step”*, but a special token concated prior to each reasoning step. It is in the token sequences, so it is not *“without visual tokens”*. Because transformers allow all tokens to attend to prior context, the Interleave Token can access:
>  - Previously selected visual tokens;
>  - The input image tokens;
>  - All prior reasoning steps.
>
> Therefore, the Interleave Token **is not blind to visual input**, but can “see” both the text reasoning steps and visual tokens before that step, to choose the related visual tokens. This process is visualized in Figure 2, where the red token marks the Interleave Token.

---

### Official Review · Reviewer_Ed2K · 2025-07-02

**Clarity:** 3
**Significance:** 2
**Originality:** 4
**Rating:** 5
**Confidence:** 5

**Summary:**

The paper aims at improving the mathematical reasoning capabilities of the VL models using the visual signals in the chain-of-thoughts. Specifically, it argues that the prior work does not focus on the fine-grained visual details (e.g., rely on box-level cues), and require external tools which come with additional inference costs and large-scale training. To resolve this, the paper proposes MINT-CoT framework. In particular, the authors (a) curate 50K problems with fine-grained visual cue annotations, (b) train a vision-language model to focus on the cues via an interleave tokens, and (c) performs progressive training (e.g., text-only, SFT, and RL) to further refine the model performance. Ultimately, the authors show that the method achieves large improvements in the performance on MathVista and GeoQA dataset.

**Questions:**

Mentioned in strengths and weaknesses

**Ethical Concerns:**

["NO or VERY MINOR ethics concerns only"]

**Final Justification:**

Rebuttal performs several important comparisons to understand the usefulness of the method.

**Limitations:**

Mentioned in strengths and weaknesses

**Quality:**

2

**Strengths And Weaknesses:**

Strengths

1. The experimental design is quite interesting. The idea behind creating a nice dataset with fine-grained visual cues will be a very usual artifact for the community.
2. The method to train an interleave token that decides the regions to focus on automatically is novel and seems to work well on MathVista and GeoQA.
3. The three-stage training pipeline is also sensible and the ablation studies showcase the usefulness of diverse design choices (Table 3-4).

Weaknesses

1. The choice of the base model as Qwen-2-VL-7B seems strange to me given Qwen-2.5-VL-7B exists in Table 1. For instance, Qwen-2.5-VL achieves 66.6% on MathVista instead of 40% Qwen-2-VL-7B. It is unclear if the method can be utilized to improve the frontier VLMs or not. I imagine that the gains will not look as large as ones claimed in the paper (30% on MathVista) once you train with Qwen-2.5-VL model.
2. The evaluation is quite narrow – just MathVista and GeoQA. Infact, GeoQA is a part of MathVista already. There are many more evaluations that are quite popular – MathVerse, MathVision, We-Math, MMMU-Pro that should be tested. There should be some general-purpose evals too like VQAv2, GQA, Blink, ChartQA etc to understand the impact of such training on overall visual capabilities of the model. In addition, it would be better to include the numbers for the entire mathvista data instead of a particular subset.
3. As suggested by Table 3 and 4, it seems that the biggest gain over the base model is from text-only CoT which is the default method nowadays. In this light, I believe that the paper overclaims its numbers. Instead of saying +34% on MathVista over the base model, I think the right baseline should be text-only CoT (default) method for reasoning training.
4. It is not clear if text-only SFT followed by text-only RL is worse than MINT-CoT framework or not. It is a valid baseline which seems missing in the current setup.

---

> ### Author Rebuttal · Authors · 2025-07-31
>
> Thank you for your thoughtful review and constructive feedback. We greatly appreciate your recognition of our dataset design, the novelty of the Interleave Token, and the effectiveness of our three-stage training pipeline. We address your concerns below.
>
>  ### 1. Choice of Qwen-2-VL-7B as Base Model
> We choose Qwen-2-VL-7B as the base model to more clearly demonstrate the effectiveness of our method, as it leaves more room for improvement and highlights relative gains. To address your concern, we also conduct additional experiments using the stronger Qwen-2.5-VL-7B model. As shown in the table below, our method still brings consistent improvements on MathVista-Math, confirming its generality across different base models.
> | Model | Base model | Text-only CoT SFT | Interleaved CoT SFT | Interleaved CoT RL |
> |-|-|-|-|-|
> | Qwen2.5-VL-7B-Instruct | 66.66      | 71.85      |  72.22    | **75.93**   |
> | Qwen2-VL-7B-Instruct  | 41.11      | 64.07            | 67.78        | **73.70**   |
>
> Due to the limited time available during the rebuttal period, we did not perform any hyperparameter tuning and directly adopted those from Qwen-2-VL-7B. Despite this, the MINT-CoT framework on Qwen-2.5-VL-7B still shows significant potential for improvement and demonstrates the robustness of our approach.
>
> ### 2. Evaluation on other benchmarks
> > The evaluation is quite narrow – just MathVista and GeoQA.
>
> We appreciate this suggestion and have extended our evaluation to cover additional datasets. Specifically:
>  - In our supplementary materials, we have added the evaluation results on the Mathematics section of **MMStar** benchmark (Table 2 in Suppl.)
>  - We did not evaluate on **MathVision**, since it was used in constructing our training data (aligned with the training data of Mulberry), which would make evaluation results biased.
>  - Due to the time limit, we add evaluation experiments on the Vision Only domain of **Mathverse**, which can well represent the improvement of our method.
>  - Moreover, we add experiments on the Mathematics section of the **MMMU-Pro-V**. Although this subset contains only 60 questions, our method still achieved the best performance.
>
> The combined results of those benchmarks are shown below:
> | Benchmark  | Qwen2-VL-7B-Instruct  | Text-only CoT SFT | Interleaved CoT SFT | Interleaved CoT RL |
> |----------------------------------|--------------------|----------------------|---------------------|----------------|
> | MathVista-Math | 41.11              | 64.07                | 67.78               | **73.70**       |
> | GeoQA                            | 37.80              | 59.02                | 62.07               | **64.72**          |
> | MMStar-Math                      | 46.4               | 67.6                 | 68.0                | **69.6**           |
> | MathVerse_MINI (Vision Only)     | 23.98        | 32.74              | 33.25            | **36.42**    |
> | MMMU_Pro_V                       | 25                 | 25                   | **30**                  | **30**             |
>
> >There should be some general-purpose evals too like VQAv2, GQA, Blink, ChartQA …
>
>
> The focus of our research is specifically for multi-modal math problems, including the interleave token for finer-grained selection and a dataset construction pipeline. These designs target math-specific challenges, such as geometric structures and compositional diagrams. Whereas our methodology can also generalize to general-purpose scenarios:
>  - Our dataset curation pipeline can be adapted to other tasks. The image gridding strategy, auxiliary OCR tools, keyword extraction, and final visual index annotation can also be applied to general visual reasoning tasks. For math reasoning tasks, we need to align fine-grained visual indices with the reasoning steps. In contrast, general visual reasoning tasks do not require such a high level of granularity, so our pipeline remains suitable for general tasks as well.
>  - The introduction of an Interleave Token for grounding reasoning steps in corresponding visual content is a general mechanism that enhances multimodal alignment and interpretability across various tasks.
>  - Our three-stage training pipeline can be easily adapted to many other general tasks.
>
> While our current work focuses on math-specific challenges, we acknowledge the importance of evaluating our method on broader vision-language tasks. Due to the domain gap between our mathematical reasoning dataset and benchmarks like VQAv2, GQA, BLINK, and ChartQA, we did not include them in our current evaluation. We plan to explore this in future work.
>
> > In addition, it would be better to include the numbers for the entire mathvista data …
>
> We chose to evaluate only the MathVista-Math subset because our method specifically targets mathematical reasoning tasks. The full MathVista dataset includes general visual question answering tasks (e.g., VQA, TQA), which fall outside the scope of our training data and method design.
>
> ### 3. Text-only CoT Baseline
> Thanks for your suggestion. We conduct experiments on the text-only SFT followed by text-only RL. As shown below, MINT-CoT still outperforms this stronger baseline:
> | Model | Qwen2-VL-7B-Instruct | Text-only CoT SFT | Interleaved CoT SFT |  Text only SFT + Text only RL | MINT-CoT-7B |
> | - | - | - | - | - | - |
> | MathVista-Math | 41.11    | 64.07                | 67.78               | 70.74 | **73.70** |
>
> This confirms that our improvements specifically arise from the interleaved visual reasoning enabled by MINT-CoT.

---

> > ### Comment · Reviewer_Ed2K · 2025-07-31
> > **Reviewer Reply**
> >
> > Hi,
> >
> > I thank the authors for including more experiments with Qwen-2.5-7B-Instruct, broader evaluation, and inclusion of Text only SFT+Text only RL baseline. With these improvements, I am increasing my score to 5.

---

> > > ### Author Response · Authors · 2025-08-02
> > >
> > > Thank you very much for acknowledging our rebuttal and efforts!

---

### Note · Authors · 2025-08-12

Dear Area Chair, Senior Area Chair, and Reviewers,

We would like to express our sincere gratitude for your invaluable time and insightful feedback, which have been instrumental in strengthening our paper.

We are honored that reviewers recognized our work's diverse strengths, acknowledging the research as **"well-motivated"** (PFDq), the design as **"interesting"** (Ed2K), the core method as **"novel"** (nxuS), and the results as **"significant"** (XQF3), while highlighting our dataset as a **"significant contribution"** (nxuS).

In response to the insightful feedback, we took the following actions to address all major concerns:
 - **Experimental Rigor & Scope** (Ed2K, PFDq): We validated our method on a stronger baseline model (Qwen-2.5-VL) and present our evaluation results on four benchmarks, demonstrating our method's significant effectiveness and generalizability.
 - **Dataset Quality** (XQF3): We conducted a human evaluation to substantiate the quality of our MINT-CoT dataset, which confirmed a high 89.49% step-wise alignment accuracy and verified its reliability.
 - **Core Mechanism** (nxuS, PFDq): We ran new targeted ablation studies to prove the superiority of visual tokens over text-only alternatives (nxuS). We also clarified a key misunderstanding of our Interleave Token's functionality (PFDq).

We were very encouraged that these efforts successfully resolved the core concerns of Reviewers Ed2K and XQF3, both of whom subsequently raised their scores. We are also confident that our new evidence and detailed responses have fully addressed the points raised by Reviewers nxuS and PFDq.

Our work, MINT-CoT, contributes the following to the community:
 - **Novelty**: It is the first framework to enable dynamic, fine-grained interleaved visual tokens at each step of a multimodal Chain-of-Thought process, fundamentally enhancing how models integrate visual evidence during reasoning.
 - **Contribution**: 1) The 54K-sample MINT-CoT dataset with token-level grounding; 2) The Interleave Token for dynamic visual selection; and 3) A three-stage pipeline that significantly boosts MLLM math reasoning.

We are confident this work represents a significant advance in multimodal reasoning. We respectfully ask that you consider our detailed responses, the positive developments during the rebuttal, and the significant value of our contributions in your final evaluation.

Thank you again for your time and consideration.

Best regards,

Paper 784 Authors

---

### Decision · Program_Chairs · 2025-09-17

**Decision:**

Accept (poster)

**Comment:**

Paper summary:  The paper present MINT-CoT a method to improve mathematical reasoning capabilities of VLMs.  The core contribution is the "interleave token mechanism" which dyanmically selects visual tokens related to the current mathematical semantics in the reasoning process and interleaves them with textual reasoning. The paper also has a curated dataset of 50K problems.  Findings include improvements in MathVista and GeoQA datasets and other results provided during the rebuttal.

Strengths/reasons to accept are: novelty of the approach, dataset contributions, and strong experimental results.
Salient weaknesses are mentioned by reviewer nxuS which the AC concurs with. The AC asks the authors to address these concerns to improve the paper.
-  "the effectiveness of this pipeline fundamentally relies on the accuracy of the selected visual tokens, which in turn depends on the perception ability of the visual encoder. ... Overall, without ensuring accurate visual token selection, the performance improvement observed from interleaving with textual tokens still lacks a clear and convincing explanation."

Summary of Review and Discussion:  the four reviewers gave scores of 3, 4, 4, 5.  The reviewer who gave "3" did not participate in the discussion and did not respond to the rebuttal -- the AC has looked at the authors' response and deems it to be satisfactory.